# DIFFUSION ALIGNMENT AS VARIATIONAL EXPECTATION-MAXIMIZATION

**Jaewoo Lee**[1,2*] **Minsu Kim**[1,3] **Sanghyeok Choi**[4] **Inhyuck Song**[1] **Sujin Yun**[5]
**Hyeongyu Kang**[1] **Woocheol Shin**[1] **Taeyoung Yun**[1] **Kiyoung Om**[1] **Jinkyoo Park**[1,6]

[1]KAIST    [2]MongooseAI    [3]Mila - Quebec AI Institute    [4]University of Edinburgh
[5]Mila, Université de Montréal    [6]Omelet

## ABSTRACT

Diffusion alignment aims to optimize diffusion models for the downstream objective. While existing methods based on reinforcement learning or direct backpropagation achieve considerable success in maximizing rewards, they often suffer from reward over-optimization amortization and mode collapse. We introduce **Diffusion Alignment as Variational Expectation-Maximization (DAV)**, a framework that formulates diffusion alignment as an iterative process alternating between two complementary phases: the E-step and the M-step. In the E-step, we employ test-time search to generate diverse and reward-aligned samples. In the M-step, we refine the diffusion model using samples discovered by the E-step. We demonstrate that DAV can optimize reward while preserving diversity for both continuous and discrete tasks: text-to-image synthesis and DNA sequence design. Our code is available at `https://github.com/Jaewoopudding/dav`.

## 1 INTRODUCTION

Diffusion models (Ho et al., 2020; Song et al., 2021) excel at generating high-fidelity samples across diverse domains, from image synthesis (Rombach et al., 2022; Ho et al., 2022), robotics (Chi et al., 2023) to computational biology (Sahoo et al., 2024). Beyond generating high-likelihood samples, many real-world applications necessitate samples optimized for external criteria; the aesthetic quality of images (Schuhmann, 2022), or the biological activity of DNA enhancers (Gosai et al., 2023).

To align the diffusion model with downstream objectives, various fine-tuning methods for diffusion models have been proposed. Prior works can be typically divided into two categories: (1) RL-based fine-tuning and (2) direct backpropagation. RL-based approaches (Fan et al., 2023; Black et al., 2023; Venkatraman et al., 2024) optimize the parameters of diffusion models with a reverse-KL objective using on-policy data. This combination is prone to mode-seeking behavior, which may cause premature convergence and mode collapse, severely degrading sample quality and diversity (Kim et al., 2025c). Direct backpropagation (Clark et al., 2023; Prabhudesai et al., 2023) attains higher sample efficiency but depends on sharp, brittle gradient signals from learned reward functions (Trabucco et al., 2021), often leading to severe over-optimization (Skalse et al., 2022). Thus, there is a pressing need for a fine-tuning framework that can effectively maximize rewards without sacrificing the diversity and naturalness of the pretrained diffusion model.

To this end, we propose **Diffusion Alignment as Variational Expectation-Maximization (DAV)**. Inspired by (Levine, 2018), our framework is based on the variational Expectation-Maximization (EM) algorithm (Neal & Hinton, 1998; Jordan et al., 1999), iteratively alternating between the E-step for exploration and the M-step for amortization. The **E-step (Exploration)** aims to discover diverse and high-reward samples from the variational posterior. To effectively capture the multimodal structure of posterior distribution, we invest additional test-time computation (Kim et al., 2025c; Zhang et al., 2025) via techniques such as gradient-based guidance (Grathwohl et al., 2021; Guo et al., 2024) and importance sampling, enabling a thorough exploration of promising regions.

The subsequent **M-step (Amortization)** updates $p_\theta$ to $p_{\theta'}$ by distilling the knowledge from the discovered samples back into the parameters of the diffusion model. Unlike conventional RL-based

---

*Correspondence to: jaewoo@kaist.ac.kr

methods that optimize a reverse-KL divergence, a *mode-seeking* objective that concentrates on a single dominant mode (Fan et al., 2023; Venkatraman et al., 2024; Uehara et al., 2024b), our M-step update corresponds to minimization of the forward-KL divergence, a *mode-covering* objective that encourages the model to cover all diverse modes discovered through the E-step (Chan et al., 2022). Therefore, the iteration of EM steps leads to a synergetic cycle, where the M-step adaptively refines the model towards a multi-modal aligned distribution, and the improved model in turn enables the E-step to gather samples from a more aligned distribution while preserving diversity.

To demonstrate the versatility of our method, we apply DAV to align diffusion models in both continuous (text-to-image synthesis) and discrete (DNA sequence design) domains. For the image synthesis task, we fine-tune the Stable Diffusion v1.5 model (Rombach et al., 2022) to optimize for given external rewards—including aesthetic quality (Schuhmann, 2022) and non-differentiable objectives like image compressibility and incompressibility (Black et al., 2023)—while mitigating the mode collapse of the images. For DNA sequence design, we fine-tune a discrete masked diffusion model (Sahoo et al., 2024) to design DNA enhancers (Gosai et al., 2023) that achieve high target activity while preserving the naturalness and diversity of generated sequences.

## 2 RELATED WORKS

### 2.1 DIFFUSION ALIGNMENT

**Fine-tuning approaches.** RL-based fine-tuning frames the denoising process as a sequential decision-making problem, optimizing a policy to maximize a black-box reward function (Black et al., 2023; Fan et al., 2023; Uehara et al., 2024b; Su et al., 2025). In parallel, direct backprop-agation approaches propagate the gradient signal from the differentiable reward function through the diffusion denoising chain, greatly improving sample efficiency (Xu et al., 2023; Clark et al., 2023; Prabhudesai et al., 2023). Despite progress, both fine-tuning methods suffer from reward over-optimization (Skalse et al., 2022), due to the mode-seeking behavior of reverse-KL optimization in reinforcement learning (Korbak et al., 2022) and the brittle gradient signal from the reward model (Kim et al., 2025b). Recently, Liu et al. (2024); Domingo-Enrich et al. (2025) propose to fine-tune the continuous diffusion model using the gradient signal of the reward function to follow the tilted distribution $p_{\text{pretrained}}(x) \cdot \exp(r(x))$, where $r(x)$ is the reward function. While effective at improving performance and mitigating over-optimization, these methods require a differentiable reward function and are not straightforward to extend to discrete diffusion models.

**Test-time inference approaches.** Test-time inference approaches allocate additional computation during generation to find aligned outputs without altering model weights. Techniques include guid-ance (Dhariwal & Nichol, 2021; Chung et al., 2023; Yu et al., 2023; Bansal et al., 2023), and test-time search methods (Ma et al., 2025; Zhang et al., 2025; Li et al., 2024b; Singhal et al., 2025; Kim et al., 2025a; Li et al., 2025; Jain et al., 2025). Despite their effectiveness in bypassing additional post-training phases, these methods face drawbacks: guidance-based approaches often suffer from underoptimization (Kim et al., 2025a), and search-based algorithms demand substantial computa-tional overhead, rendering them impractical for real-world applications.

DAV unifies the strengths of these two paradigms by leveraging a test-time search to collect diverse, aligned samples and then distill the gathered knowledge back into the model through a principled Expectation-Maximization algorithm. By amortizing test-time search into the parameters of the dif-fusion model, DAV achieves strong alignment and sampling diversity without extensive computation requirements at inference time. While recent methods (Liu et al., 2024; Domingo-Enrich et al., 2025) focus on continuous diffusion models and rely on the differentiability of the reward function, DAV naturally extends to both continuous and discrete diffusion without requiring any assumption on the differentiability of the reward function, making it a more general and widely applicable framework.

## 3 BACKGROUNDS

### 3.1 DIFFUSION MODELS

Diffusion models are generative models that learn data distributions by adding noise to data and then training how to reverse the noising process (Sohl-Dickstein et al., 2015). They comprise two

opposing Markov chains, a forward process $q$ that gradually adds noise to the data, and a learned reverse process $p_\theta$ that denoises the state to recover the original data:

$$q(x_{1:T} \mid x_0) = \prod_{t=1}^{T} q(x_t \mid x_{t-1}), \qquad p_\theta(x_{0:T}) = p_T(x_T) \prod_{t=1}^{T} p_\theta(x_{t-1} \mid x_t),$$

where $x_0 \sim q_{\text{data}}$ is clean data, $x_T$ is pure noise, and $p_T(x_T)$ is a simple base distribution from which we can easily sample. Depending on the data modality, we can parameterize diffusion processes with a Gaussian distribution for continuous diffusion or a Categorical distribution for discrete diffusion. The details are elaborated in Appendix G.

The reverse process $p_\theta$ is learned by optimizing the evidence lower bound (ELBO) on the data log-likelihood $\mathbb{E}_{x_0 \sim q_{\text{data}}}[\log p_\theta(x_0)]$, which can be decomposed as follows:

$$\mathbb{E}_q[\log p_\theta(x_0|x_1)] - \sum_{t=2}^{T} \mathbb{E}_q[D_{\text{KL}}(q(x_{t-1}|x_t, x_0) \parallel p_\theta(x_{t-1}|x_t))] - D_{\text{KL}}(q(x_T|x_0) \parallel p_T(x_T)).$$

## 3.2 Markov Decision Process

We formulate the fine-tuning of the reverse diffusion process as a Markov Decision Process (MDP). Following prior works (Fan & Lee, 2023; Black et al., 2023), we define a finite-horizon MDP with sparse reward and deterministic transitions, denoted as the tuple $(\mathcal{S}, \mathcal{A}, P, r, \gamma, \rho_0)$, where $\mathcal{S}$ is the state space, $\mathcal{A}$ the action space, $P$ the transition dynamics, $\gamma \subset [0, 1]$ the discount factor, $r : \mathcal{S} \times \mathcal{A} \to \mathbb{R}$ the reward function, and $\rho_0$ the initial state distribution. Unlike previous approaches, we incorporate the discount factor $\gamma$. Within this MDP, we want to fine-tune our diffusion policy $\pi$. The specific formulations are as follows:

$$s_t \triangleq (x_{T-t}, T - t) \quad \pi_\theta(a_t|s_t) \triangleq p_\theta(x_{T-t-1}|x_{T-t}) \quad P(s_{t+1}|s_t, a_t) \triangleq \delta_{(x_{T-t-1}, T-t-1)}$$

$$a_t \triangleq x_{T-t-1} \qquad \rho_0(s_0) \triangleq (p_T, \delta_T) \qquad r(s_t, a_t) \triangleq \begin{cases} R(x_0) & \text{if } t = T - 1 \\ 0 & \text{otherwise.} \end{cases}$$

$R(x_0)$ is an external reward function defined on the clean data space. For brevity, we mainly use notations with $x_t$'s (rather than $(s_t, a_t)$) in the subsequence sections. Also, with a slight abuse of notation, we often denote $r(x_t, x_{t-1})$ instead of $r((x_t, t), x_{t-1})$.

## 3.3 KL-Divergence Regularized Reinforcement Learning

Following Abdolmaleki et al. (2018); Wu et al. (2019); Kumar et al. (2019), we consider maximization of the KL-regularized RL objective for a given diffusion model $p_{\text{prior}} = p_\theta$, i.e.,

$$p_{\theta^*} = \arg\max_{p_{\theta'}} \mathbb{E}_{\tau \sim p_{\text{prior}}} \left[ \sum_{t=1}^{T} \gamma^{T-t} (r(x_t, x_{t-1}) - \alpha D_{\text{KL}}(p_{\theta'}(x_{t-1}|x_t) \| p_{\text{prior}}(x_{t-1}|x_t))) \right], \quad (1)$$

where $\tau = (x_T, x_{T-1}, \ldots, x_0)$ is a trajectory sampled from $\rho_0$ and $p_{\text{prior}}$, and $\alpha > 0$ controls the strength of the regularization. We define soft Q-function of state-action pair $(x_t, x_{t-1})$ as follows:

$$Q^*_{\text{soft}}(x_t, x_{t-1}) = r(x_t, x_{t-1})$$
$$+ \mathbb{E}_{\tau \sim p_{\theta^*}} \left[ \sum_{s=1}^{t-1} \gamma^{t-s} \left( r(x_s, x_{s-1}) - \alpha D_{\text{KL}}(p_{\theta^*}(\cdot|x_s) \| p_{\text{prior}}(\cdot|x_s)) \right) \middle| x_t, x_{t-1} \right]$$
$$(2)$$

Following Uehara et al. (2024a), KL-regularized soft Bellman equations are given by:

$$V^*_{\text{soft}}(x_t) = \alpha \log \mathbb{E}_{x_{t-1} \sim p_{\text{prior}}(\cdot|x_t)} \left[ \exp\left( \tfrac{1}{\alpha} Q^*_{\text{soft}}(x_t, x_{t-1}) \right) \right], \quad (3)$$

$$Q^*_{\text{soft}}(x_t, x_{t-1}) = r(x_t, x_{t-1}) + \gamma \cdot \mathbb{E}_{x_{t-1} \sim p_{\text{prior}}(\cdot|x_t)} \left[ V^*_{\text{soft}}(x_{t-1}) \right]. \quad (4)$$

Under the discounted diffusion MDP with sparse reward and deterministic transition of Section 3.2, we can set terminal conditions $V^*_{\text{soft}}(x_0) = 0$, $Q^*_{\text{soft}}(x_1, x_0) = r(x_0)$, and approximate the soft Q-function by using Tweedie's formula (Efron, 2011; Li et al., 2024b) as follows:

$$Q^*_{\text{soft}}(x_t, x_{t-1}) \approx \gamma^{t-1} r(\hat{x}_0(x_{t-1})), \quad (5)$$

where $\hat{x}_0(x_t) = \mathbb{E}_{x_0 \sim p_{\text{prior}}}[x_0|x_t]$ denotes the approximated posterior mean. (See Appendix B for details.)

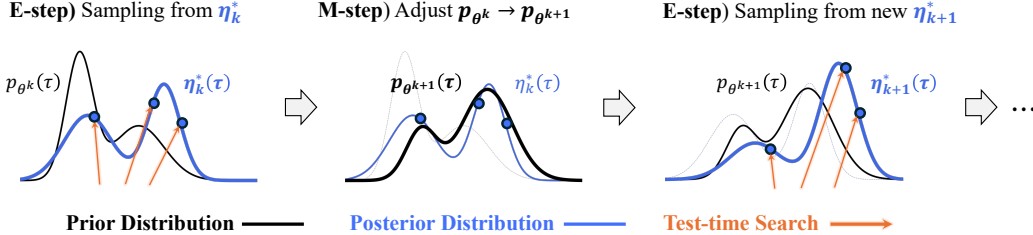

Figure 1: Conceptual illustration of DAV. DAV alternates between E-step, where trajectories are obtained via test-time search, and M-step, where the diffusion model parameters $\theta$ are updated by amortizing the posterior into the policy. By iterating these two steps, DAV progressively refines the diffusion model toward a multi-modal aligned distribution.

# 4 DIFFUSION ALIGNMENT AS VARIATIONAL EM

In this section, we introduce a detailed mechanism of **Diffusion Alignment as Variational Expectation-Maximization (DAV)**. As illustrated in Figure 1, DAV aligns diffusion models by iteratively alternating between an E-step and an M-step. The E-step involves a test-time search guided by a soft Q-function to effectively discover high-reward, multi-modal trajectories from the variational posterior distribution. Subsequently, in the M-step, the diffusion model is updated by distilling the information from searched trajectories through the E-step, progressively learning to generate outputs that are aligned with the reward function. The training procedure of our method is summarized in the pseudo code provided in Appendix A.

## 4.1 VARIATIONAL EXPECTATION-MAXIMIZATION FORMULATION

Inspired by Levine (2018), we cast diffusion alignment as maximizing the likelihood of a binary optimality variable $\mathcal{O}$, defined as $p_\theta(\mathcal{O} = 1|\tau) \propto \exp(\sum_{t=1}^{T} r_t(x_t, x_{t-1})/\alpha)$. This directly implies that the alignment objective is to maximize the marginal likelihood of optimality as follows:

$$\max_\theta \log p_\theta(\mathcal{O} = 1).$$

Importantly, $\theta$ does not directly parameterize this marginal likelihood, but instead induces a distribution over trajectories by $p_\theta(\tau) = p_T(x_T) \prod_{t=1}^{T} p_\theta(x_{t-1}|x_t)$. In other words, since $\theta$ specifies the diffusion reverse process, alignment can be formulated by introducing a latent trajectory variable $\tau$ that bridges between the model parameters and the optimality variable. The reverse process yields a trajectory $\tau$, which contains $x_0$ that is subsequently evaluated by the reward function. Hence, $\tau$ acts as an unobserved latent variable connecting $\theta$ to the observed optimality outcome $\mathcal{O}$. The resulting incomplete log-likelihood is expressed as $\log p_\theta(\mathcal{O} = 1) = \log \int p_\theta(\tau, \mathcal{O} = 1)d\tau$.

Directly maximizing the incomplete log-likelihood is intractable due to the hierarchical structure of denoising trajectories. Instead, we introduce a variational distribution $\eta(\tau) = p_T(x_T) \prod_{t=1}^{T} \eta(x_{t-1} \mid x_t)$, to approximate the intractable posterior $p_\theta(\tau|\mathcal{O} = 1)$ and convert the marginal likelihood optimization into a tractable variational inference problem as follows (See Appendix C for details):

$$\log p_\theta(\mathcal{O} = 1) \geq \mathbb{E}_{\tau \sim \eta}\left[\sum_{t=1}^{T}\left(\frac{r(x_t, x_{t-1})}{\alpha} + \log \frac{p_\theta(x_{t-1} \mid x_t)}{\eta(x_{t-1} \mid x_t)}\right)\right] = \mathcal{J}_\alpha(\eta, p_\theta), \qquad (6)$$

**Introducing discount factor.** The high stochasticity of the diffusion reverse process makes early denoising steps less impactful on the final outcome (Ho et al., 2020), necessitating a discounting mechanism to attenuate credit assignment of $x_t$ with large timestep $t$.

**Proposition 1** (Lower bound on likelihood of $\mathcal{O}$ with discount factor $\gamma$). *Let $\gamma \in (0, 1]$ be the discount factor. The likelihood of the optimality variable $\mathcal{O}$ admits the following lower bound:*

$$\mathcal{J}_{\alpha,\gamma}(\eta, p_\theta) \triangleq \mathbb{E}_{\tau \sim \eta(\tau)}\left[\sum_{t=1}^{T} \gamma^{T-t}\left(\frac{r(x_t, x_{t-1})}{\alpha} + \log \frac{p_\theta(x_{t-1} \mid x_t)}{\eta(x_{t-1} \mid x_t)}\right)\right]. \qquad (7)$$

*Proof.* See Appendix D.

Optimizing the ELBO, $\mathcal{J}_{\alpha,\gamma}(\eta, p_\theta)$, is often approached with an EM-based RL algorithm, alternating between the E-step (posterior inference) and the M-step (model update) (Dayan & Hinton, 1997). However, prior EM-based RL approaches exhibit a critical weakness in the E-step. They approximate the posterior by reweighting on-policy samples or past experiences from a replay buffer (Peters & Schaal, 2007; Abdolmaleki et al., 2018; Nair et al., 2020). However, if the behavioral policy deviates significantly from the posterior distribution $p_\theta(\tau|\mathcal{O} = 1)$, this approach critically misspecifies the posterior, guiding the M-step toward a biased and suboptimal distribution.

Therefore, we redesign the EM-based RL for diffusion alignment. Let $\theta^k$ be the model parameter at the $k$-th iteration. In the E-step, we first determine the posterior distribution that maximizes the ELBO, i.e., $\eta_k^*(\tau) = \arg\max_\eta \mathcal{J}_{\alpha,\gamma}(\eta, p_{\theta^k})$. Then, we employ test-time search (Singhal et al., 2025; Kim et al., 2025c) to obtain approximate samples that follow $\eta_k^*(\tau)$. In the M-step, we maximize the ELBO by updating the model parameter, $\theta^{k+1} = \arg\max_\theta \mathcal{J}_{\alpha,\gamma}(\eta_k^*, p_\theta)$, which corresponds to minimizing the forward KL divergence using the samples from the $\eta_k^*$.

## 4.2 E-STEP: TEST-TIME SEARCH FOR POSTERIOR INFERENCE

In the E-step, we sample trajectories from the variational posterior distribution $\eta_k^*$ by employing test-time search. We first determine $\eta_k^*(\tau) = \arg\max_\eta \mathcal{J}_{\alpha,\gamma}(\eta, p_{\theta^k})$. Since our objective is equivalent to the KL-regularized RL objective in Equation (1), the optimal variational distribution, $\eta_k^*(\tau)$, is the production of the soft optimal policy that takes the form of a reward-tilted distribution:

$$\eta_k^*(x_{t-1}|x_t) \propto p_{\theta^k}(x_{t-1}|x_t)\exp(Q_{\text{soft},\theta^k}^*(x_t, x_{t-1})/\alpha). \qquad (8)$$

The derivation is detailed in Section E.1. However, directly sampling from $\eta_k^*$ is intractable. To overcome this, we approximate it using a two-stage local search to generate the subsequent state, $x_{t-1}$. The first stage of this search involves sampling a set of candidate particles. Specifically, we draw M intermediate particles, $\{x_{t-1}^m\}_{m=1}^M$, from a proposal distribution, $\hat{\eta}_k$. If the gradient signal of the reward is available, we can construct an effective proposal distribution by using gradient guidance (Grathwohl et al., 2021; Kim et al., 2025a). Subsequently, we refine the intermediate particles via importance sampling, which effectively pushes the samples closer to $\eta_k^*$ (Li et al., 2024b). We detail our search procedure in Section E.2.

Note that the test-time search in DAV is a modular component. This design allows for any algorithm capable of approximating the target posterior distribution to be substituted into the E-step. Therefore, DAV is not tied to a specific search technique and can directly benefit from future advancements in test-time search methods (Jain et al., 2025; Zhang et al., 2025; Yoon et al., 2025).

## 4.3 M-STEP: AMORTIZING TEST-TIME SEARCH INTO DIFFUSION MODELS

In the M-step, we update $\theta$ by distilling trajectories from the E-step. This corresponds to maximizing ELBO by projecting $\eta_k^*$ onto the diffusion $p_\theta$ via forward KL minimization on searched trajectories:

$$\theta^{k+1} = \arg\max_\theta \mathcal{J}_{\alpha,\gamma}(\eta_k^*(\tau), \theta) = \arg\min_\theta D_{\text{KL}}(\eta_k^*(\tau)||p_\theta(\tau)). \qquad (9)$$

Minimizing this forward KL divergence is equivalent to maximizing the log-likelihood of the trajectories from $\eta_k^*$. Since the policy update is performed via gradient ascent rather than an exact maximization, it constitutes a partial M-step in a Generalized EM framework (Dempster et al., 1977). Nevertheless, as long as the gradient ascent increases the ELBO, the monotonic improvement property of EM is preserved. The training objective for **DAV** is thus:

$$\mathcal{L}_{\text{DAV}}(\theta) = \mathbb{E}_{\tau \sim \eta_k^*}[-\log p_\theta(\tau)] = \mathbb{E}_{\tau \sim \eta_k^*}\left[\sum_{t=1}^T -\log p_\theta(x_{t-1}|x_t)\right]. \qquad (10)$$

To prevent the capability loss of the pretrained model, we introduce **DAV-KL**, which adds a KL-divergence term to penalize deviation from the initial pretrained policy $p_{\theta^0}$:

$$\mathcal{L}_{\text{DAV-KL}}(\theta) = \mathbb{E}_{\tau \sim \eta_k^*}\left[\sum_{t=1}^T -\log p_\theta(x_{t-1}|x_t)\right] + \lambda D_{\text{KL}}(p_\theta(x_{t-1}|x_t)||p_{\theta^0}(x_{t-1}|x_t)). \qquad (11)$$

where the coefficient $\lambda$ controls the trade-off between aligning with the expert policy $\eta_k^*$ and preserving the knowledge of the pretrained model.

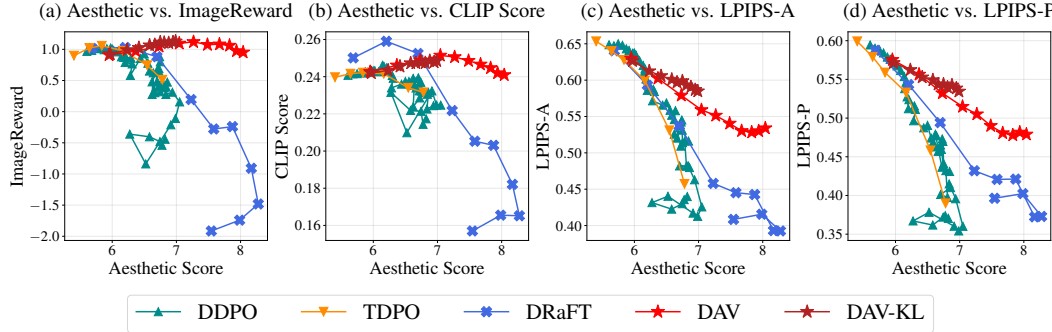

Figure 2: Training dynamics of our methods and baselines, with performance marked every 10 epochs. All methods were trained for 100 epochs, except for DDPO, which was trained for 500 epochs. Our approaches successfully preserve alignment score and diversity compared to baselines.

| Method | Aesthetic ($\uparrow$) | LPIPS-A ($\uparrow$) | ImageReward ($\uparrow$) |
|---|---|---|---|
| Pretrained | 5.40 (0.01) | **0.65 (0.01)** | 0.90 (0.01) |
| DDPO | 6.83 (0.16) | 0.48 (0.05) | 0.27 (1.01) |
| TDPO | 6.78 (0.28) | 0.39 (0.10) | 0.51 (0.47) |
| DRaFT | 7.22 (0.22) | 0.46 (0.04) | 0.19 (0.64) |
| DAV | 8.04 (0.07) | 0.53 (0.03) | 0.95 (0.21) |
| DAV-KL | 6.99 (0.04) | 0.58 (0.01) | 1.13 (0.04) |
| DAS | 7.22 (0.01) | 0.65 (0.01) | 1.07 (0.03) |
| DAV Posterior | **9.18 (0.07)** | 0.53 (0.01) | 0.91 (0.21) |
| DAV-KL Posterior | 8.66 (0.09) | 0.58 (0.01) | **1.14 (0.08)** |

Table 1: Comparison on text-to-image synthesis benchmarks. All results are reported as mean with standard deviation in parentheses. Our methods and TDPO are shown at the 100th epoch, while DDPO and DRaFT are taken from the last checkpoint before over-optimization. Methods above the midline are fine-tuning approaches, and those below are test-time search methods.

## 5 EXPERIMENTS

This section empirically evaluates the ability of our framework to optimize rewards while preserving sample diversity and naturalness in continuous and discrete diffusion. We demonstrate the versatility of our framework across two distinct data modalities: text-to-image synthesis using a latent diffusion model (Rombach et al., 2022) and DNA sequence design via a discrete diffusion model (Sahoo et al., 2024). We denote the amortized policies as DAV and DAV-KL, and their posterior samples, obtained through test-time inference as explained in Section 4.2, as DAV Posterior and DAV-KL Posterior, respectively. Implementation details and hyperparameter settings for all experiments are provided in Appendix H. To reflect the versatility of DAV, our baselines mainly consist of methods that are agnostic to data modality and reward function differentiability. We also compare our approach with a representative test-time search method, as it is the core mechanism of the E-step.

### 5.1 CONTINUOUS DIFFUSION: TEXT-TO-IMAGE SYNTHESIS

We use Stable Diffusion v1.5 (Rombach et al., 2022) as our base pretrained model across all text-to-image experiments. We employ a set of 40 simple animal prompts for fine-tuning as provided by Kim et al. (2025c). All results are averaged over three random seeds.

#### 5.1.1 EXPERIMENTAL SETUP

**Metrics.** We use the differentiable LAION aesthetic score (Schuhmann, 2022) as our primary reward. We evaluate the results for two key failure modes: reward over-optimization and diversity collapse. To detect over-optimization, we measure prompt alignment using CLIPScore (Radford et al., 2021) for semantic consistency and ImageReward (Xu et al., 2023) for human preference. Sample

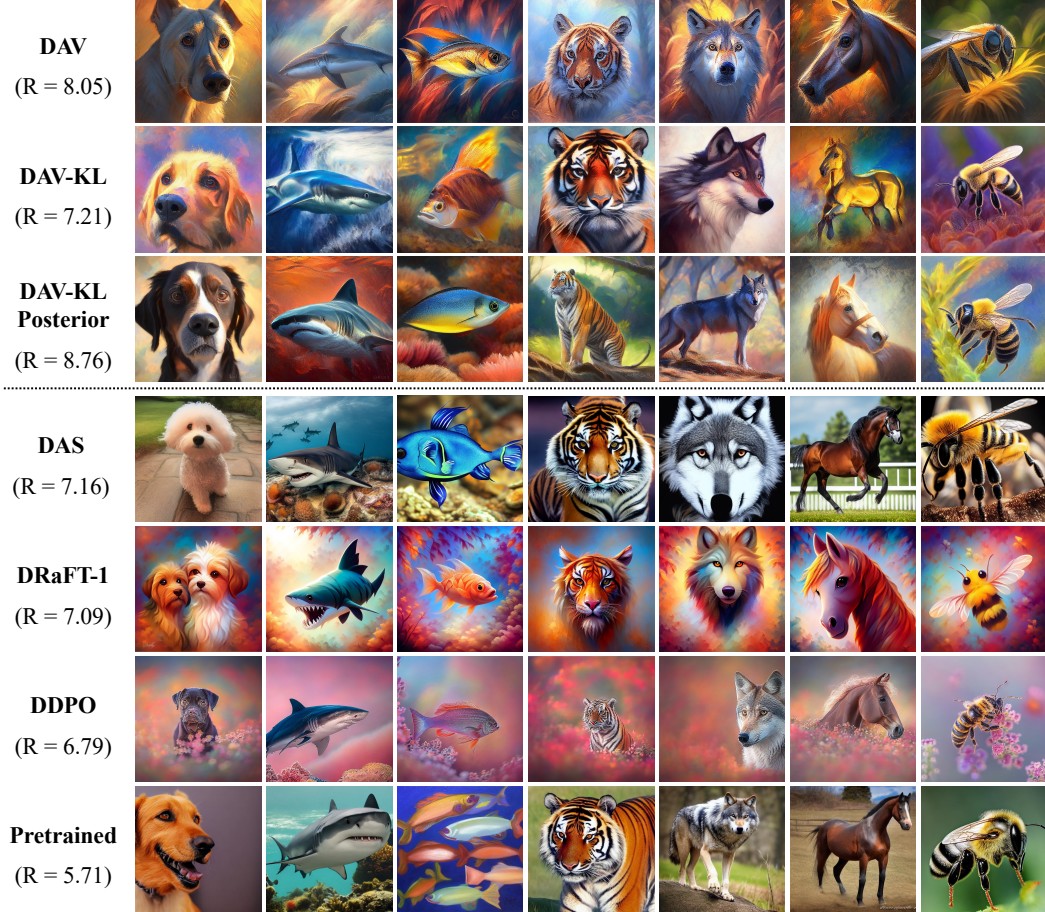

Figure 3: Qualitative comparison of our methods with DAS, DRaFT-1, DDPO, and pretrained model. Results for our methods are reported after 100 epochs of fine-tuning. For DDPO and DRaFT, we sample images from the last checkpoint before significant collapse. All images in the figure are collected from multiple runs, and the score reported under each method is the average aesthetic score of the seven sampled images.

diversity is quantified using LPIPS (Zhang et al., 2018), reporting the average distance across all samples (LPIPS-A) and within samples from the same prompt (LPIPS-P).

**Baselines.** We compare DAV against representative baselines include DDPO (Black et al., 2023) for RL-based fine-tuning, DRaFT (Clark et al., 2023) for direct backpropagation, and DAS (Kim et al., 2025c) for test-time search. We also include TDPO (Zhang et al., 2024b), a gradient-free RL-based method specifically designed to mitigate reward over-optimization.

### 5.1.2 RESULTS

Figure 2 shows the training dynamics for each method, demonstrating how alignment and diversity metrics evolve as the aesthetic reward is optimized. The results in Figure 2-(a) and Figure 2-(b) indicate that DAV and DAV-KL maintain a high alignment score while the baselines exhibit a sharp degradation in alignment scores. Similarly, Figure 2-(c) and Figure 2-(d) demonstrate that DAV and DAV-KL are substantially better at preserving sample diversity throughout the fine-tuning process.

Table 1 provides a quantitative comparison of our methods against the baselines. The first key observation is a comparison between our methods and the fine-tuning baselines. DAV achieves a significantly higher reward (8.04) than both DDPO (6.83) and DRaFT (7.22), while maintaining a high ImageReward score (0.95) comparable to the pretrained model. In DAV-KL, KL-regularization induces a trade-off: it enhances diversity and ImageReward while incurring a reduction in reward.

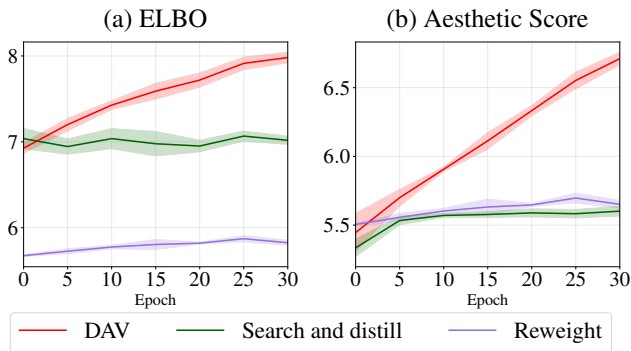

Figure 4: Comparison of ELBO and aesthetic score trends for DAV and its ablated baselines.

As illustrated in Figure 3, both DAV and DAV-KL generate well-aligned samples without producing the repetitive backgrounds seen in the outputs of DDPO and DRaFT-1.

The second key observation from Table 1 is the comparison among test-time search methods. DAV Posterior achieves the highest aesthetic score (9.18), substantially outperforming DAS (7.22), while DAV-KL Posterior obtains the best ImageReward (1.14) with competitive aesthetic performance (8.66). Although our methods exhibit a slight decrease in diversity compared to DAS, our methods deliver clear gains in reward and alignment quality.

**Effect of E-step Variants in DAV.** In *Search and distill*, the model is updated using samples from $\eta_0^*$ instead of $\eta_k^*$, skipping the update of the posterior distribution. In *Reweight*, we replace test-time search with trajectory reweighting by exponentiated reward, following Abdolmaleki et al. (2018). By ablating key components of the E-step, we evaluate the effectiveness of test-time search in their ability to optimize the ELBO.

Figure 4 illustrates the ELBO and the corresponding aesthetic score trends for each variation. We empirically validated that DAV consistently improves ELBO, although we cannot guarantee monotonic improvement due to approximation errors in the E-step. The ablated baselines not only fail to optimize ELBO, as expected, but also fail to increase the aesthetic score. These results suggest the importance of test-time computation in the E-step for our variational EM framework.

**Additional analysis.** We conduct sensitivity analyses of the hyperparameters of DAV in Appendix I. Lower $\alpha$ increases reward but risks over-optimization, while higher values improve diversity. Setting $\gamma^T \approx 0$ stabilizes optimization by limiting early-step credit assignment. We analyze the effect of the number of distillation steps in the M-step and the number of particles in importance sampling. We also evaluated DAV with non-differentiable objectives, such as the compressibility and incompressibility rewards from Black et al. (2023), and we report the results in Appendix J.

## 5.2 Discrete Diffusion: DNA Sequence Design

We pretrained the masked discrete diffusion model (Sahoo et al., 2024) on the large-scale DNA enhancer dataset from Gosai et al. (2023). The dataset consists of 700k DNA sequences (200-bp length) and their corresponding enhancer activity in human cell lines, as measured by massively parallel reporter assays (MPRAs). All subsequent results are averaged over six random seeds.

### 5.2.1 Experimental Setup

**Metrics.** Our metric setup follows the methodology of Wang et al. (2025). As the target reward, we use a trained Enformer network (Avsec et al., 2021) to predict enhancer activity (Pred-Activity ($\uparrow$)). To avoid data leakage, we train two distinct Enformer models on disjoint splits of the enhancer dataset (Gosai et al., 2023): one provides rewards during fine-tuning, and the other serves solely as a held-out evaluator. We further assess the generated sequences on three criteria: diversity, naturalness, and biological validity. We quantify diversity using the average Levenshtein distance ($\uparrow$) (Haldar & Mukhopadhyay, 2011) between generated sequences. We assess naturalness using the 3-mer Pearson Correlation (3-mer Corr ($\uparrow$)) against the top 0.1% most active enhancers in the dataset. Finally, to measure biological validity and detect over-optimization, we use an independently trained classifier for Chromatin Accessibility (ATAC-Acc ($\uparrow$)) (Consortium et al., 2012; Lal et al., 2024).

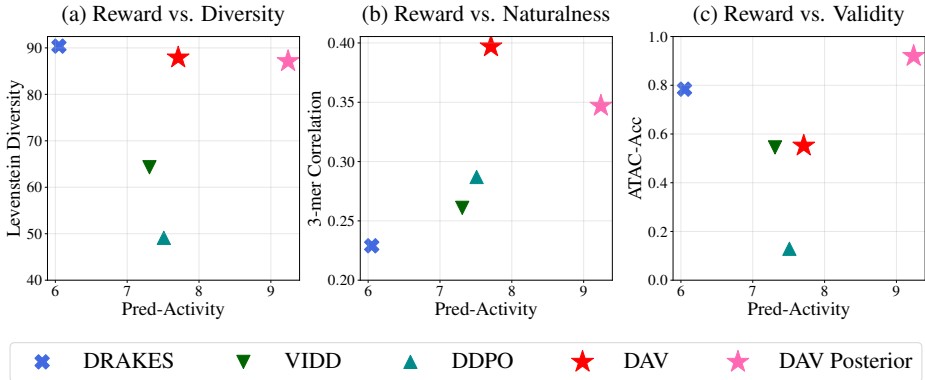

Figure 5: Performance comparison of DAV and baseline models. The x-axis represents the reward (Pred-Activity), while the y-axis shows (a) Diversity (Levenshtein Diversity), (b) Naturalness (3-mer Corr), and (c) Validity (ATAC-Acc).

**Baselines.** We compare DAV against representative baselines of discrete diffusion model alignment. For direct backpropagation on discrete models, our baseline is DRAKES (Wang et al., 2025). For RL-based fine-tuning, we compare against DDPO (Black et al., 2023) and VIDD (Su et al., 2025).

### 5.2.2 RESULTS

As shown in Figure 5, our methods outperform baselines in generating enhancer DNA sequences. In Figure 5-(a), DAV achieves the best trade-off between reward and diversity. Figure 5-(b) shows that our methods achieve the best performance in both reward and naturalness. Figure 5-(c) further confirms that our methods effectively balance reward and validity. Moreover, DAV Posterior further enhances reward optimization while maintaining diversity and naturalness. These results showcase the versatility of DAV in the discrete diffusion alignment, where it effectively optimizes for reward while maintaining high sample diversity. The entire table of results is presented in Table 2.

## 6 DISCUSSION

**Conclusion.** We introduced DAV, a novel diffusion alignment framework based on the variational Expectation-Maximization algorithm. The E-step performs test-time search guided by a soft Q-function to discover high-reward, diverse, and natural samples; the M-step amortizes the result of test-time search by forward-KL distillation, fine-tuning diffusion models to the high-reward distribution while preserving the diversity and naturalness. We validated DAV on two distinct data modalities: continuous diffusion for image synthesis and discrete diffusion for DNA sequence design. Our results show that DAV effectively fine-tunes diffusion models to optimize rewards while mitigating over-optimization and diversity collapse in both domains.

**Limitations and future works.** Our work presents two primary avenues for future research. The first addresses the main limitation of our current framework: the computational overhead of the test-time search in the E-step. Fortunately, the E-step is modular, allowing for the direct integration of more efficient search algorithms. We anticipate that leveraging recent and future advances in test-time search (Zhang et al., 2025; Li et al., 2025) will substantially mitigate this bottleneck. The second limitation is the approximation error in the soft Q-function, which may guide the test-time search toward suboptimal distributions. This error primarily stems from the inaccuracy of Tweedie's formula for approximating the posterior mean at $x_t$ with high noise levels (Chung et al., 2023). Future work could address this by employing distillation techniques (Salimans & Ho, 2022; Song et al., 2023). Distillation techniques substantially reduce the number of denoising steps required for accurate $x_0$ prediction, expected to enhance posterior mean approximation at large timesteps and yield more reliable Q-function approximations.

## ACKNOWLEDGEMENT

We thank the anonymous reviewers for their insightful comments and suggestions, which significantly improve our manuscript. This work is supported by the National Research Foundation of Korea (NRF) grant funded by the Korea government (MSIT) (No. RS-2024-00410082), and NRF grant funded by the Korea government(MSIT) (No. RS-2025-00563763).

## THE USE OF LARGE LANGUAGE MODELS

Large Language Models were employed exclusively for two auxiliary tasks: (1) minor polishing of the manuscript text for improving grammar and readability, and (2) limited assistance in code implementation for debugging syntax or refactoring functions. Importantly, LLMs did not contribute to the conception of the research problem, the development of the core methodology, or the design and execution of experiments. All critical ideas, methods, and analyses presented in this paper are the original work of the authors.

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

# A    PSEUDO CODE OF DAV

---

**Algorithm 1** Diffusion Alignment as Variational EM (DAV)

---

**Require:** Pretrained diffusion parameters $\theta^0$.
1: Initialize model parameters $\theta \leftarrow \theta^0$
2: **for** $k = 1, \ldots, N$ **do**
3:      Initialize dataset of trajectories $\mathcal{D} \leftarrow \emptyset$
4:      ***E-step: Posterior Exploration via Test-Time Search***
5:      **for** $b = 1, \ldots, B$ **do**
6:          Sample initial noise $x_T \sim \rho_0(x_T)$
7:          Initialize trajectory $\tau \leftarrow \{x_T\}$
8:          **for** $t = T, \ldots, 1$ **do**
9:              Sample $M$ particles from $\hat{q}_k(\cdot|x_t)$ with Equation (56) or Equation (59)
10:              Compute weights $\{w^m\}_{m=1}^M$ with Equation (62)
11:              Resample $x_{t-1} \sim \text{Categorical}(\{w^m\}_{m=1}^M)$
12:              Append $x_{t-1}$ to trajectory $\tau$
13:          **end for**
14:          Add completed trajectory $\tau$ to $\mathcal{D}$
15:      **end for**
16:
17:      ***M-step: Amortization via Forward KL Projection***
18:      **for** $\tau \in \mathcal{D}_k$ **do**
19:          Update $\theta$ by minimizing Equation (10) or Equation (11)
20:      **end for**
21: **end for**
22: **return** Optimized parameters $\theta^N$

---

# B  APPROXIMATION OF THE SOFT Q-FUNCTION

In KL-regularized RL for diffusion fine-tuning, prior works approximate the Soft Q-function for the undiscounted MDP case where $\gamma = 1$ (Uehara et al., 2024a; Li et al., 2024b). By leveraging the recursive soft Bellman equation and the posterior mean approximation by Tweedie's formula (Efron, 2011; Chung et al., 2023), the soft Q-function can be approximated as:

$$Q^*_{\text{soft}}(x_t, x_{t-1}; \gamma = 1) \approx r(\hat{x}_0(x_t)) \tag{12}$$

However, this approximation does not hold in the more general discounted setting where $\gamma \neq 1$. We find that for a discounted MDP, the recursive form of the Bellman equation instead yields lower and upper bounds on the Soft Q-function. For $t \geq 2$ and $0 < \gamma < 1$, we derive the following inequalities:

$$\alpha\gamma \log \mathbb{E}_{x_{0:t-1} \sim p_{\text{prior}}} \left[ \exp\left( \frac{\gamma^{t-2} r(x_0)}{\alpha} \right) \right] \leq Q^*_{\text{soft}}(x_t, x_{t-1}) \leq \alpha\gamma^{t-1} \log\left( \mathbb{E}_{p_{\text{prior}}} \left[ \exp\left( \frac{r(x_0)}{\alpha} \right) \right] \right).$$

For inducing upper inequalities, we start by substituting the definition of the soft Q-function into the soft value function, and we obtain the recursive expression of the soft value function:

$$V^*_{\text{soft}}(x_t) = \alpha \log \mathbb{E}_{x_{t-1} \sim p_{\text{prior}}(\cdot|x_t)} \left[ \exp\left( \frac{r(x_t, x_{t-1})}{\alpha} + \frac{\gamma}{\alpha} V^*_{\text{soft}}(x_{t-1}) \right) \right] \tag{13}$$

Based on the sparse reward MDP defined in Section 3.2, the immediate reward $r(x_t, x_{t-1})$ is zero for all steps $t \geq 2$. This simplifies the recursion to

$$V^*_{\text{soft}}(x_t) = \alpha \log \mathbb{E}_{x_{t-1} \sim p_{\text{prior}}(\cdot|x_t)} \left[ \exp\left( \frac{1}{\alpha} V^*_{\text{soft}}(x_{t-1}) \right) \right], \quad \text{for } t \geq 2. \tag{14}$$

With the terminal condition $V^*_{\text{soft}}(x_0) = 0$, the soft Q-value at $t = 1$ is simply the immediate reward, $Q^*_{\text{soft}}(x_1, x_0) = r(x_1, x_0) = r(x_0)$. By substituting this into the definition of the soft value function, we can derive the value at $t = 1$ and subsequently the Q-function at $t = 2$ as:

$$V^*_{\text{soft}}(x_1) = \alpha \log \mathbb{E}_{x_0 \sim p_{\text{prior}}(\cdot|x_1)}[\exp(r(x_0))/\alpha] \tag{15}$$

$$Q^*_{\text{soft}}(x_2, x_1) = \gamma\alpha \log \mathbb{E}_{x_0 \sim p_{\text{prior}}(\cdot|x_1)} \left[ \exp\left( \frac{r(x_0)}{\alpha} \right) \right] \tag{16}$$

To simplify the notation, we define the auxilary variable $\beta_t(x_t) := \exp\left( V^*_{\text{soft}}(x_t)/\alpha \right)$. From the definition of $V^*_{\text{soft}}$, the case at $t = 1$ is $\beta_1(x_1) = \mathbb{E}_{x_0 \sim p_{\text{prior}}}[\exp(r(x_0)/\alpha)]$. Then the recursion of soft Bellman equations for $t \geq 2$ can be written as:

$$\beta_t(x_t) = \mathbb{E}_{x_{t-1} \sim p_{\text{prior}}}[\beta_{t-1}(x_{t-1})^\gamma]. \tag{17}$$

Since the function $z \mapsto z^\gamma$ is concave for $0 < \gamma < 1$, applying Jensen's inequality yields the following relationship: $(\mathbb{E}[Z])^\gamma \geq \mathbb{E}[Z^\gamma]$. By iteratively applying this result to the recursive definition of $\beta_t(x_t)$ ), we can establish a lower bound as follows:

$$\beta_t(x_t) = \mathbb{E}_{x_{t-1} \sim p_{\text{prior}}}[\beta_{t-1}(x_{t-1})^\gamma] \tag{18}$$

$$= \mathbb{E}_{x_{t-1} \sim p_{\text{prior}}}[(\mathbb{E}_{x_{t-2} \sim p_{\text{prior}}}[\beta_{t-2}(x_{t-2})^\gamma])^\gamma] \tag{19}$$

$$\geq \mathbb{E}_{x_{t-2:t-1} \sim p_{\text{prior}}}[(\beta_{t-2}(x_{t-2}))^{\gamma^2}] \tag{20}$$

$$\geq \mathbb{E}_{x_{t-3:t-1} \sim p_{\text{prior}}}[(\beta_{t-3}(x_{t-3}))^{\gamma^3}] \tag{21}$$

$$\cdots \tag{22}$$

$$\geq \mathbb{E}_{x_{1:t-1} \sim p_{\text{prior}}}[(\beta_1(x_1))^{\gamma^{t-1}}] \tag{23}$$

$$= \mathbb{E}_{x_{1:t-1} \sim p_{\text{prior}}}[\mathbb{E}_{x_0 \sim p_{\text{prior}}}[\exp(r(x_0)/\alpha)]^{\gamma^{t-1}}] \tag{24}$$

By applying Jensen's inequality again, we can obtain the lower bound of the $\beta_t(x_t)$:

$$\mathbb{E}_{x_{0:t-1} \sim p_{\text{prior}}}[(\exp(r(x_0)/\alpha)^{\gamma^{t-1}}] \leq \beta_t(x_t) \tag{25}$$

In a similar manner, the upper bound of $\beta_t(x_t)$ for $t \geq 2$ is derived using $\mathbb{E}[Z^\gamma] \leq (\mathbb{E}[Z])^\gamma$:

$$\beta(x_t) = \mathbb{E}_{x_{t-1} \sim p_{\text{prior}}}[\beta_{t-1}(x_{t-1})^\gamma] \tag{26}$$

$$\leq (\mathbb{E}_{x_{t-1} \sim p_{\text{prior}}}[\beta_{t-1}(x_{t-1})])^\gamma \tag{27}$$

$$\leq (\mathbb{E}_{x_{t-1}, x_{t-2} \sim p_{\text{prior}}}[\beta_{t-2}(x_{t-2})])^{\gamma^2} \tag{28}$$

$$\cdots \tag{29}$$

$$\leq (\mathbb{E}_{x_{1:t-1} \sim p_{\text{prior}}}[\beta_1(x_1)])^{\gamma^{t-1}} \tag{30}$$

$$= (\mathbb{E}_{x_{1:t-1} \sim p_{\text{prior}}}[\mathbb{E}_{x_0 \sim p_{\text{prior}}}[\exp(r(x_0)/\alpha)]])^{\gamma^{t-1}} \tag{31}$$

By the law of total expectation, the nested expectations can be combined into a single expectation over the trajectory $x_{0:t-1}$, which yields the final upper bound for $\beta_t(x_t)$:

$$\beta_t(x_t) \leq (\mathbb{E}_{x_{0:t-1} \sim p_{\text{prior}}}[\exp(r(x_0)/\alpha)])^{\gamma^{t-1}} \tag{32}$$

Combining these results, the bounds on $\beta_t$ for $t \geq 2$ is summarized as:

$$\mathbb{E}_{x_{0:t-1} \sim p_{\text{prior}}}[(\exp(r(x_0)/\alpha))^{\gamma^{t-1}}] \leq \beta_t(x_t) \leq \mathbb{E}_{x_{0:t-1} \sim p_{\text{prior}}}[\exp(r(x_0)/\alpha)]^{\gamma^{t-1}} \tag{33}$$

By substituting $\beta_t(x_t) = \exp(V^*_{\text{soft}}(x_t)/\alpha)$ back into the inequality, we obtain the bounds on the exponentiated soft value function:

$$\mathbb{E}_{x_{0:t-1} \sim p_{\text{prior}}}\left[\exp\left(\frac{\gamma^{t-1} r(x_0)}{\alpha}\right)\right] \leq \exp\left(\frac{V^*_{\text{soft}}(x_t)}{\alpha}\right) \leq \left(\mathbb{E}_{x_{0:t-1} \sim p_{\text{prior}}}\left[\exp\left(\frac{r(x_0)}{\alpha}\right)\right]\right)^{\gamma^{t-1}} \tag{34}$$

Taking the logarithm of all parts and multiplying by $\alpha$ yields the bounds on the soft value function $V^*_{\text{soft}}(x_t)$:

$$\alpha \log \mathbb{E}_{x_{0:t-1} \sim p_{\text{prior}}}\left[\exp\left(\frac{\gamma^{t-1} r(x_0)}{\alpha}\right)\right] \leq V^*_{\text{soft}}(x_t) \leq \alpha \gamma^{t-1} \log \left(\mathbb{E}_{x_{0:t-1} \sim p_{\text{prior}}}\left[\exp\left(\frac{r(x_0)}{\alpha}\right)\right]\right). \tag{35}$$

Using the relation $Q^*_{\text{soft}}(x_t, x_{t-1}) = \gamma V^{*\text{soft}}(x_{t-1})$ for $t \geq 3$ from Equation (4), we can derive the corresponding bounds for the soft Q-function:

$$\alpha \gamma \log \mathbb{E}_{x_{0:t-1} \sim p_{\text{prior}}}\left[\exp\left(\frac{\gamma^{t-2} r(x_0)}{\alpha}\right)\right] \leq Q^*_{\text{soft}}(x_t, x_{t-1}) \leq \alpha \gamma^{t-1} \log \left(\mathbb{E}_{p_{\text{prior}}}\left[\exp\left(\frac{r(x_0)}{\alpha}\right)\right]\right).$$

A remarkable simplification occurs when we approximate the log-sum-exp terms in both the lower and upper bounds using Tweedie's formula (Efron, 2011; Chung et al., 2023). We find that both complex bounds are approximated to the same simple, closed-form expression: $\gamma^{t-1} r(\hat{x}_0(x_{t-1}))$. This result, which is also consistent with the exact solutions for the boundary cases at $t = 1$ and $t = 2$, yields the following unified, first-order approximation for the soft Q-function:

$$Q^*_{\text{soft}}(x_t, x_{t-1}) \approx \gamma^{t-1} r(\hat{x}_0(x_{t-1})). \tag{36}$$

## C  DERIVATION OF EVIDENCE LOWER BOUND

We derive the ELBO of the marginal log-likelihood of the optimality variable, $\log p_\theta(\mathcal{O} = 1)$. The derivation begins by introducing a variational distribution $\eta(\tau)$ over the trajectories. By applying Jensen's inequality, we can derive the ELBO as follows:

$$\log p_\theta(\mathcal{O} = 1) = \log \int p_\theta(\tau)p(\mathcal{O} = 1|\tau)\, d\tau \tag{37}$$

$$= \log \int \eta(\tau)\frac{p_\theta(\tau)p(\mathcal{O} = 1|\tau)}{\eta(\tau)}\, d\tau \tag{38}$$

$$\geq \int \eta(\tau)\left(\log p(\mathcal{O} = 1|\tau) + \log \frac{p_\theta(\tau)}{\eta(\tau)}\right) d\tau \tag{39}$$

$$\propto \mathbb{E}_{\tau \sim \eta}\left[\sum_{t=1}^{T}\left(\frac{r(x_t, x_{t-1})}{\alpha} + \log \frac{p_\theta(x_{t-1}|x_t)}{\eta(x_{t-1}|x_t)}\right)\right] =: J_\alpha(\eta, p_\theta) \tag{40}$$

This final expression constitutes the ELBO, which serves as our tractable surrogate objective for alignment.

## D  PROOF OF PROPOSITION 1

**Proposition 1** (Lower bound on likelihood of $\mathcal{O}$ with discount factor $\gamma$). *Let $\gamma \in (0, 1]$ be the discount factor. The likelihood of the optimality variable $\mathcal{O}$ admits the following lower bound:*

$$\mathcal{J}_{\alpha,\gamma}(\eta, p_\theta) \triangleq \mathbb{E}_{\tau \sim \eta(\tau)} \left[ \sum_{t=1}^{T} \gamma^{T-t} \left( \frac{r(x_t, x_{t-1})}{\alpha} + \log \frac{p_\theta(x_{t-1} \mid x_t)}{\eta(x_{t-1} \mid x_t)} \right) \right]. \tag{7}$$

*Proof.* For clarity, we stick to MDP notations in Section 3.2. First, as suggested in Levine (2018), we introduce an absorbing state $s_{\text{absorb}}$ to take into account the discount factor $\gamma$ and let $\bar{\mathcal{S}} := \mathcal{S} \cup \{s_{\text{absorb}}\}$. Each action $a$ at $s$ make transition to $s' \in \mathcal{S}$ following the original dynamics $P$ with probability $\gamma$ and to $s_{\text{absorb}}$ with probability $1 - \gamma$, i.e., the modified transition probability $\bar{P}$ becomes:

$$\bar{P}(s' \mid s \in \mathcal{S}, a) = \begin{cases} \gamma P(s' \mid s, a) & \text{if } s' \in \mathcal{S} \\ 1 - \gamma & \text{if } s' = s_{\text{absorb}}, \end{cases} \tag{41}$$

and $\bar{P}(s_{\text{absorb}} \mid s_{\text{absorb}}, a) = 1$.

Similarly, the modified reward becomes:

$$\bar{r}(s, a) = \begin{cases} r(s, a) & \text{if } s \in \mathcal{S} \\ 0 & \text{if } s = s_{\text{absorb}}. \end{cases} \tag{42}$$

Also, $\bar{p}_\theta$ and $\bar{\eta}$ define distributions over trajectories in the modified MDP, with $\bar{p}_\theta(\cdot|s_{\text{absorb}}) = \bar{\eta}(\cdot|s_{\text{absorb}}) = \delta_{s_{\text{absorb}}}(\cdot)$ and $\bar{p}_\theta(\cdot|s) = p_\theta(\cdot|s)$, $\bar{\eta}(\cdot|s) = \eta(\cdot|s)$ if $s \in \mathcal{S}$.

We can derive the ELBO for the modified MDP in the same way as Equation (40), which gives:

$$\log p_\theta(\mathcal{O} = 1) \geq \mathbb{E}_{\tau \sim \bar{\eta}, \bar{P}} \left[ \sum_{t=0}^{T-1} \left( \frac{\bar{r}(s_t, a_t)}{\alpha} + \log \frac{\bar{p}_\theta(a_t|s_t)}{\bar{\eta}(a_t|s_t)} \right) \right] =: \mathcal{J}_{\alpha,\gamma}(\bar{\eta}, \bar{p}_\theta) \tag{43}$$

By linearity of expectation, we can swap the sum and the expectation:

$$\mathcal{J}_{\alpha,\gamma}(\bar{\eta}, \bar{p}_\theta) = \sum_{t=0}^{T-1} \mathbb{E}_{(s_t, a_t) \sim \bar{\eta}, \bar{P}} \left[ \frac{\bar{r}(s_t, a_t)}{\alpha} + \log \frac{\bar{p}_\theta(a_t|s_t)}{\bar{\eta}(a_t|s_t)} \right]. \tag{44}$$

Consider the event $E_t$, meaning the agent has not yet reached the absorbing state up to time $t$, i.e., $s_t \in \mathcal{S}, \forall t \in [0, 1, \ldots, t]$ and $\Pr(E_t) = \gamma^t$. Recall the law of total expectation:

$$\mathbb{E}_{(s_t, a_t) \sim \bar{\eta}, \bar{P}} \left[ \cdot \right] = \Pr(E_t) \cdot \mathbb{E}_{(s_t, a_t) \sim \bar{\eta}, \bar{P}} \left[ \cdot | E_t \right] + (1 - \Pr(E_t)) \cdot \mathbb{E}_{(s_t, a_t) \sim \bar{\eta}, \bar{P}} \left[ \cdot | \neg E_t \right].$$

If $\neg E_t$, then $s_t = s_{\text{absorb}}$, $\bar{r}(s_t, a_t) = 0$, and $\log \frac{\bar{p}_\theta(a_t|s_t)}{\bar{\eta}(a_t|s_t)} = 0$. On the other hand, when conditioned on $E_t$, $\bar{\eta}(\cdot|s_t) = \eta(\cdot|s_t)$, $\bar{p}_\theta(\cdot|s_t) = p_\theta(\cdot|s_t)$, $\bar{r}(s_t, a_t) = r(s_t, a_t)$, and $\mathbb{E}_{(s_t, a_t) \sim \bar{\eta}, \bar{P}} \left[ \cdot | E_t \right] = \mathbb{E}_{(s_t, a_t) \sim \eta} \left[ \cdot \right]$. Now, we can rewrite $\mathcal{J}_{\alpha,\gamma}$ in the original MDP:

$$\mathcal{J}_{\alpha,\gamma}(\bar{\eta}, \bar{p}_\theta) = \sum_{t=0}^{T-1} \Pr(E_t) \cdot \mathbb{E}_{(s_t, a_t) \sim \bar{\eta}, \bar{P}} \left[ \frac{\bar{r}(s_t, a_t)}{\alpha} + \log \frac{\bar{p}_\theta(a_t|s_t)}{\bar{\eta}(a_t|s_t)} \bigg| E_t \right] \tag{45}$$

$$= \sum_{t=0}^{T-1} \gamma^t \mathbb{E}_{(s_t, a_t) \sim \eta} \left[ \frac{r(s_t, a_t)}{\alpha} + \log \frac{p_\theta(a_t|s_t)}{\eta(a_t|s_t)} \right] \tag{46}$$

$$= \mathbb{E}_{\tau \sim \eta} \left[ \sum_{t=0}^{T-1} \gamma^t \left( \frac{r(s_t, a_t)}{\alpha} + \log \frac{p_\theta(a_t|s_t)}{\eta(a_t|s_t)} \right) \right]. \tag{47}$$

By rewriting the last line using notations with $x_t$'s:

$$\mathcal{J}_{\alpha,\gamma}(\eta, p_\theta) = \mathbb{E}_{\tau \sim \eta} \left[ \sum_{t=1}^{T} \gamma^{T-t} \left( \frac{r(x_t, x_{t-1})}{\alpha} + \log \frac{p_\theta(x_{t-1}|x_t)}{\eta(x_{t-1}|x_t)} \right) \right]. \tag{48}$$

$\square$

# E    DETAILS OF TEST-TIME SEARCH FOR THE E-STEP

This section details the test-time search procedure for DAV. In the E-step, we are required to sample trajectories from the optimal posterior distribution, $\eta^*(\tau)$. We first simplify the problem by defining an optimal policy, $\eta^*(x_{t-1}|x_t)$, and decompose the optimal posterior distribution as $\eta^*(\tau) = \eta_T(x_T) \prod_{t=1}^{T} \eta^*(x_{t-1}|x_t)$. We then draw samples from this policy using a two-stage process: (1) Constructing proposal distribution $\hat{\eta}(x_{t-1}|x_t)$ using gradient guidance from the reward function (Grathwohl et al., 2021; Dhariwal & Nichol, 2021), and (2) Employing importance sampling to refine these samples, correcting for mismatches with the true posterior.

## E.1    DERIVING SOFT OPTIMAL POLICY FOR EXPLORATION

For the fixed policy parameters of diffusion model at the $k$-th iteration, $p_{\theta^k}$, the optimal posterior policy $\eta_k^*(x_{t-1}|x_t)$ for $1 \leq t \leq T$ is defined as the policy that maximizes the regularized objective $\mathcal{J}_{\alpha,\gamma}(\eta, p_{\theta^k})$. Because our objective $\mathcal{J}_{\alpha,\gamma}(\eta, p_{\theta^k})$ is equivalent to the KL-regularized RL objective in Equation (1), $\eta_k^*(x_{t-1}|x_t)$ is the soft optimal policy. Let $Q_{\text{soft},\theta^k}^*$ denote the soft Q-function for the objective regularized with respect to the prior policy $p_{\theta^k}$. Then, the soft optimal policy is found by solving the following maximization problem:

$$\eta_k^*(x_{t-1}|x_t) = \underset{\eta(\cdot|x_t)}{\arg\max} \, \mathbb{E}_{x_{t-1} \sim \eta(\cdot|x_t)} \left[ \frac{Q_{\text{soft},\theta^k}^*(x_t, x_{t-1})}{\alpha} - \log \frac{\eta(x_{t-1}|x_t)}{p_{\theta^k}(x_{t-1}|x_t)} \right] \tag{49}$$

$$= \underset{\eta(\cdot|x_t)}{\arg\max} \, \mathbb{E}_{x_{t-1} \sim \eta(\cdot|x_t)} \left[ \log \frac{p_{\theta^k}(x_{t-1}|x_t) \exp(Q_{\text{soft},\theta^k}^*(x_t, x_{t-1})/\alpha)}{\eta(x_{t-1}|x_t)} \right] \tag{50}$$

$$= \underset{\eta(\cdot|x_t)}{\arg\min} \, D_{\text{KL}}(\eta(x_{t-1}|x_t) || \frac{1}{Z} p_{\theta^k}(x_{t-1}|x_t) \exp(Q_{\text{soft},\theta^k}^*(x_t, x_{t-1})/\alpha)) - \log Z \tag{51}$$

$$= \frac{1}{Z} p_{\theta^k}(x_{t-1}|x_t) \exp(Q_{\text{soft},\theta^k}^*(x_t, x_{t-1})/\alpha), \tag{52}$$

where $Z = \int p_{\theta^k}(x_{t-1}|x_t) \exp(Q_{\text{soft},\theta^k}^*(x_{t-1}|x_t)/\alpha) dx_{t-1}$. To summarize, the soft optimal policy $\eta_k^*(x_{t-1}|x_t)$ takes the form of a Boltzmann distribution, where the prior policy $p_{\theta^k}$ is re-weighted by the exponentiated soft Q-function to favor actions with higher expected soft returns.

## E.2    GRADIENT-GUIDED TEST-TIME SEARCH WITH IMPORTANCE SAMPLING CORRECTION

We now detail the two-stage procedure for sampling from the soft optimal policy, $\eta_k^*(x_{t-1}|x_t)$, which is designed to generate high-reward samples while preserving the diversity of the prior model:

- **Proposal construction**: If the reward function is differentiable, construct a proposal distribution $\hat{\eta}_k$ for $\eta_k^*$ using a first-order Taylor expansion.

- **Importance sampling correction**: We employ importance sampling to correct for the distributional mismatch between our proposal distribution $\hat{\eta}_k$ and the optimal policy $\eta_k^*$.

### E.2.1    CONSTRUCTING PROPOSAL DISTRIBUTION VIA TAYLOR EXPANSION

For complex reward functions where the target posterior deviates significantly from the current policy, leveraging the gradient of a differentiable reward function can dramatically improve the efficiency of the E-step. This gradient-guided exploration provides a more direct path to discovering high-reward samples.

**Continuous diffusion.**    In the case of continuous diffusion models, we incorporate the reward gradient into the reverse process by leveraging a first-order Taylor expansion, as previously suggested by Dhariwal & Nichol (2021); Kim et al. (2025c). We derive our proposal distribution from the

optimal soft policy $\eta_k^*$ by first approximating the soft Q-function with the reward function:

$$\eta_k^*(x_{t-1}|x_t) = \frac{1}{Z} p_{\theta^k}(x_{t-1}|x_t) \exp\left(\frac{1}{\alpha} Q_{\text{soft},\theta^k}^*(x_t, x_{t-1})\right) \tag{53}$$

$$\approx \frac{1}{Z} p_{\theta^k}(x_{t-1}|x_t) \exp\left(\frac{\gamma^{t-1}}{\alpha} r(\hat{x}_0(x_{t-1}))\right) \tag{54}$$

where $\hat{x}_0(x_{t-1}) = \mathbb{E}_{p_{\theta^k}}[x_0|x_{t-1}]$ and $p_{\theta^k}(x_{t-1}|x_t) = \mathcal{N}(x_{t-1}; \mu_{\theta^k}(x_t, t), \sigma_t^2 I)$. To maintain a tractable Gaussian form, we approximate $r(\hat{x}_0(x_{t-1}))$ with a first-order Taylor expansion around the mean of the prior distribution, $\mu_{\theta^k}$:

$$r(\hat{x}_0(x_{t-1})) \approx r(\hat{x}_0(x_t)) + \nabla_{x_t} r(\hat{x}_0(x_t))^T (x_{t-1} - \mu_{\theta^k}(x_t, t)). \tag{55}$$

Since $r(\hat{x}_0(x_t))$ does not depend on $x_{t-1}$, it is absorbed into $Z$, leaving only the linear term in the exponent. This process effectively incorporates the linearized reward gradient as a guidance term, shifting the mean of the prior distribution (Dhariwal & Nichol, 2021). We therefore define our proposal policy $\hat{\eta}_k$ as this modified Gaussian:

$$\hat{\eta}_k(x_{t-1}|x_t) := \frac{1}{Z} p_{\theta^k}(x_{t-1} \mid x_t) \exp\left(\frac{\gamma^{t-1}}{\alpha} r(\hat{x}_0(x_{t-1}))\right) \tag{56}$$

$$= \mathcal{N}\left(x_{t-1}; \mu_{\theta^k}(x_t, t) + \frac{\sigma_t^2}{\alpha} \gamma^{t-1} \nabla_{x_t} r(\hat{x}_0(x_t)), \sigma_t^2 I\right). \tag{57}$$

**Discrete diffusion.**   Drawing inspiration from Grathwohl et al. (2021) and Nisonoff et al. (2025), we extend gradient-guided E-step to a discrete diffusion model by incorporating Taylor expansion-based gradient exploitation of a differentiable soft optimal Q-function. Similar to the derivation process of proposal distribution of the continuous diffusion, our derivation begins with the approximated optimal policy, where the soft Q-function is approximated as a posterior mean approximation:

$$\hat{\eta}_k(x_{t-1} \mid x_t) := \frac{1}{Z} p_{\theta^k}(x_{t-1} \mid x_t) \exp\left(\frac{\gamma^{t-1}}{\alpha} r(\hat{x}_0(x_{t-1}))\right) \tag{58}$$

To create a tractable gradient signal, we apply a first-order Taylor expansion to the reward function $r(\hat{x}_0(x_{t-1}))$ around the current state estimate $\hat{x}_0(x_t)$. After substituting the expansion, we can simplify the expression. Terms that are constant with respect to $x_{t-1}$ are absorbed into the normalization constant Z. This isolates the influence of the gradient signal as a linear term that directly modifies the log-probabilities of the prior distribution $p_{\theta^k}$.

$$\hat{\eta}_k(x_{t-1} \mid x_t) \propto p_{\theta^k}(x_{t-1} \mid x_t) \exp\left(\frac{\gamma^{t-1}}{\alpha} \left[r(\hat{x}_0(x_t)) + \nabla_{x_t} r(\hat{x}_0(x_t))^\top (x_{t-1} - x_t)\right]\right) \tag{59}$$

$$\propto p_{\theta^k}(x_{t-1} \mid x_t) \exp\left(\frac{\gamma^{t-1}}{\alpha} \nabla_{x_t} r(\hat{x}_0(x_t))^\top x_{t-1}\right) \tag{60}$$

The final expression defines a new categorical distribution whose log-probabilities are those of the prior policy, shifted by a linear guidance term. Finally, the proposal distribution is Categorical with logit-additive guidance:

$$\hat{\eta}_k(x_{t-1} \mid x_t) = \prod_{\ell=1}^{L} \text{Cat}\left(x_{t-1}^\ell; \text{softmax}(\mathbf{z}_\ell)\right) \tag{61}$$

where the logits vector $\mathbf{z}_\ell \in \mathbb{R}^K$ for position $\ell$ has components:

$$[\mathbf{z}_\ell]_i = \log \pi_{\theta^k}(x_{t-1,\ell} = i \mid x_t) + \frac{\gamma^t}{\alpha} \left[\nabla_{x_{t,\ell}} r(\hat{x}_0(x_t))\right]_i,$$

and $\ell \in \{1, \cdots, L\}$: dimension, $i \in \{1, \cdots, K\}$: category index.

### E.2.2 IMPORTANCE SAMPLING CORRECTION

Since the proposal distribution $\hat{\eta}_k$ is constructed via Taylor expansion of the reward function, the resulting samples may deviate from the true optimal distribution $q_k^*$. To correct for this mismatch, we employ importance sampling. Given $M$ samples $\{x_{t-1}^m\}_{m=1}^M$ from the proposal distribution $\hat{\eta}_k$, we assign importance weight to each particle as:

$$w_{t-1}^m := \frac{\eta^*(x_{t-1}^m|x_t)}{\hat{\eta}(x_{t-1}^m|x_t)} \tag{62}$$

$$= \frac{p_{\theta^k}(x_{t-1}^m|x_t)}{\hat{\eta}(x_{t-1}^m|x_t)} \exp(\frac{1}{\alpha} Q_{\text{soft},\theta^k}^*(x_t, x_{t-1}^m)) \tag{63}$$

$$\approx \frac{p_{\theta^k}(x_{t-1}^m|x_t)}{\hat{\eta}(x_{t-1}^m|x_t)} \exp(\frac{\gamma^{t-1}}{\alpha} r(\hat{x}_0(x_{t-1}^m))) \tag{64}$$

A corrected sample from $q_k^*$ is then obtained by resampling according to the normalized weights:

$$x_{t-1} \sim \text{Cat}\left(\left\{\frac{w_{t-1}^m}{\sum_{n=1}^M w_{t-1}^n}\right\}_{m=1}^M\right) \tag{65}$$

# F EXTENDED RELATED WORKS

## F.1 SELF-TRAINING

The self-training paradigm, pioneered by the AlphaGo series (Silver et al., 2016; 2017a;b), alternates between an expert rollout stage to generate high-quality data and a distillation stage to fine-tune the model on that data via maximum likelihood estimation (MLE). This framework has been successfully adapted to Large Language Models in methods like STaR (Zelikman et al., 2024), which bootstraps reasoning from self-generated rationales, and ReST (Gulcehre et al., 2023; Singh et al., 2024; Zhang et al., 2024a), which distills a policy from a filtered set of high-quality outputs. In the same vein, DAV translates the self-training paradigm to diffusion models. Within the DAV framework, the test-time search (E-step) acts as the expert rollout to discover diverse, high-reward samples, which are then used for distillation (M-step) via MLE. To our knowledge, DAV is the first to apply the self-training framework to the problem of aligning diffusion models.

## F.2 GRPO AND DPO BASED GENERATIVE MODEL ALIGNMENT

Recently, Group Relative Policy Optimization (GRPO) has emerged as a strong method for aligning LLMs, often outperforming PPO-based approaches (Shao et al., 2024). Following this trend, several GRPO-based alignment methods for diffusion fine-tuning have been introduced, achieving strong performance in reward optimization (Liu et al., 2025; Xue et al., 2025b;a). However, their primary objective is reward maximization rather than preventing over-optimization. DAV is designed to mitigate over-optimization, providing a balanced trade-off between reward optimization, diversity, and naturalness.

There is also a line of work aligning diffusion models using Direct Policy Optimization (DPO; Rafailov et al., 2023). While these methods achieve strong human preference win rates over pre-trained models (Wallace et al., 2024; Li et al., 2024a; Zhu et al., 2025), they are fundamentally constrained by the quality of pre-collected preference datasets, which often limits their performance frontier (Xue et al., 2025b). Moreover, prior GRPO and DPO-based approaches focus primarily on visual generative tasks, whereas DAV provides a unified framework that is empirically validated on both continuous and discrete diffusion models.

# G ADDITIONAL BACKGROUNDS

## G.1 CONTINUOUS DIFFUSION

Diffusion models (Sohl-Dickstein et al., 2015; Ho et al., 2020) are a class of hierarchical generative models that learn to approximate the data distribution. A denoising diffusion model generates a sample $x_0$ by a Markov generative process—often referred to as the reverse process—that starts from a standard Gaussian prior $p_T(x_T) = \mathcal{N}(0, \mathbf{I})$:

$$p_\theta(x_{0:T}) = p_T(x_T) \prod_{t=1}^{T} p_\theta(x_{t-1} \mid x_t), \quad p_\theta(x_{t-1} \mid x_t) = \mathcal{N}(x_{t-1}; \mu_\theta(x_t, t), \sigma_t^2 I).$$

The forward process is defined by a fixed Markov chain that gradually corrupts the data $x_0$ with Gaussian noise according to a variance schedule $\{\beta_t\}_{t=1}^{T}$:

$$q(x_{1:T} \mid x_0) = \prod_{t=1}^{T} q(x_t \mid x_{t-1}), \qquad q(x_t \mid x_{t-1}) = \mathcal{N}(x_t; \sqrt{1 - \beta_t}\, x_{t-1}, \beta_t I).$$

A useful property of the forward process is that $x_t$ can be obtained directly in closed form without simulating the entire chain:

$$q(x_t \mid x_0) = \mathcal{N}(x_t; \sqrt{\bar{\alpha}_t}\, x_0, (1 - \bar{\alpha}_t)I),$$

where $\alpha_t = 1 - \beta_t$, $\bar{\alpha}_t = \prod_{s=1}^{t} \alpha_s$.

Training a diffusion model is performed by optimizing the evidence lower bound(ELBO) on the data log-likelihood $\mathbb{E}_{x_0 \sim q_{\text{data}}} [\log p_\theta(x_0)]$, which can be decomposed as follows:

$$\mathbb{E}_q[\log p_\theta(x_0|x_1)] - \sum_{t=2}^{T} \mathbb{E}_q[D_{\text{KL}}(q(x_{t-1}|x_t, x_0) \parallel p_\theta(x_{t-1}|x_t))] - D_{\text{KL}}(q(x_T|x_0) \parallel p_T(x_T)).$$

## G.2 DISCRETE DIFFUSION

Our discrete diffusion model follows the framework defined in Masked Diffusion Language Models (MDLM) (Sahoo et al., 2024).

**Notation.** We begin by considering a single discrete variable $x_0 \in \{1, 2, ..., K\}$ that belongs to a finite vocabulary of size $K$. We denote the mask token as $[M]$, which serves as an absorbing state in our forward diffusion process. The diffusion process operates over $T$ discrete time steps, with $t \in \{1, 2, ..., T\}$. We use $\mathbf{Q}_t \in [0, 1]^{K \times K}$ to denote row-stochastic transition matrices where each row sums to 1. The noise schedule is parameterized by $\alpha_t = e^{-\sigma(t)}$ where $\sigma(t) : [0, 1] \to \mathbb{R}_+$, and $\beta_t = 1 - \alpha_t$ controls the corruption rate.

**Forward diffusion process.** The forward diffusion process gradually corrupts the clean data $x_0$ through a markov chain over $T$ discrete time steps:

$$q(x_{1:T} \mid x_0) = \prod_{t=1}^{T} q(x_t \mid x_{t-1}).$$

Each forward step follows a categorical distribution:

$$q(x_t \mid x_{t-1}) = \text{Cat}(x_t; \mathbf{e}_{x_{t-1}} \mathbf{Q}_t),$$

where $\mathbf{e}_{x_{t-1}}$ is the one-hot encoding of $x_{t-1}$. The choice of $\mathbf{Q}_t$ determines the corruption process. Austin et al. (2021) deals with various options for $\mathbf{Q}_t$ such as uniform, absorbing, and Gaussian; however, we are mainly considering an absorbing kernel as in Sahoo et al. (2024):

$$(\mathbf{Q}_t)_{i,j} = \begin{cases} \beta_t, & \text{if } j = [M] \text{ and } i \neq [M] \\ 1, & \text{if } i = j = [M] \\ 1 - \beta_t, & \text{if } j = i \neq [M] \\ 0. & \text{otherwise} \end{cases}$$

Following prior works, log-linear schedule $\sigma(t) = -\log(1-t)$ gives us $\alpha_t = 1 - t$ and $\beta_t = t$, meaning the corruption probability increases linearly with time.

Additionally, marginal forward distribution can be computed as:

$$q\left(x_t \mid x_0\right) = \operatorname{Cat}\left(x_t; \mathbf{e}_{x_0} \overline{\mathbf{Q}}_t\right),$$

where $\overline{\mathbf{Q}}_t = \mathbf{Q}_1 \mathbf{Q}_2 \cdots \mathbf{Q}_t$ is the cumulative transition matrix. This analytical tractability enables sample-efficient training as we can directly sample $x_t$ from $x_0$ without simulating the entire forward chain. Let $\bar{\alpha}_t = \prod_{s=1}^{t} \alpha_s = \prod_{s=1}^{t} (1 - \beta_s)$. Then:

$$(\overline{\mathbf{Q}}_t)_{i,j} = \begin{cases} 1 - \overline{\alpha}_t, & \text{if } j = [M] \text{ and } i \neq [M] \\ 1, & \text{if } j = i = [M] \\ \overline{\alpha}_t, & \text{if } j = i \neq [M] \\ 0. & \text{otherwise} \end{cases}$$

This means the marginal distribution simplifies to:

$$q(x_t|x_0) = \begin{cases} \overline{\alpha}_t, & \text{if } x_t = x_0 \neq [M] \\ 1 - \overline{\alpha}_t, & \text{if } x_t = [M] \text{ and } x_0 \neq [M] \\ 1, & \text{if } x_0 = [M] \\ 0. & \text{otherwise} \end{cases}$$

In other words, each unmasked token $x_0$ remains unchanged with probability $\overline{\alpha_t}$. This simple binary choice makes sampling and likelihood computation extremely efficient.

**Reverse diffusion process.** Austin et al. (2021) introduces the $x_0$-parameterization for the reverse process. Sahoo et al. (2024) further simplifies the process by introducing substitution-based (SUBS) parameterization. SUBS-parameterization simplifies the reverse diffusion process by explicitly preventing the model from predicting the absorbing state ($[M]$) when conditioned on an intermediate sample, and also by ensuring that once a token is denoised, it is permanently fixed and cannot revert to a masked state in subsequent sampling steps. The reverse process in this paper is parameterized as:

$$p_\theta(x_s|x_t) = q(x_s|x_t, x_0 = \hat{x}_0(x_t, t; \theta)) = \begin{cases} \operatorname{Cat}(x_s; x_t), & \text{if } x_t \neq [M] \\ \operatorname{Cat}(x_s; \frac{(1-\alpha_s)\mathbf{m} + (\alpha_s - \alpha_t)\hat{x}_0(x_t, t; \theta)}{1 - \alpha_t} & \text{if } x_t = [M] \end{cases}$$

where the $p_\theta(x_s|x_t)$ for $s < t$ denoises from time $t$ back to time $s$, and $\mathbf{m}$ denotes the one-hot encoding of the mask token $[M]$.

**Loss function.** From the original loss in Ho et al. (2020), Sahoo et al. (2024) derives a simplified training objective through Rao-Blackwellization of the discrete-time ELBO. We define a sequence of time step pairs $(s(1), t(1)), ..., (s(T), t(T))$ where $0 \leq s(i) < t(i) \leq T$ for all $i$, which partitions the diffusion trajectory into segments. The key result is:

$$\mathcal{L}_{\text{MDLM}} = \sum_{i=1}^{T} \mathbb{E}_q \left[ D_{\text{KL}} \left( q\left(x_{s(i)} \mid x_{t(i)}, x_0\right) \| p_\theta\left(x_{s(i)} \mid x_{t(i)}\right) \right) \right]$$

$$= \sum_{i=1}^{T} \mathbb{E}_q \left[ \frac{\alpha_{t(i)} - \alpha_{s(i)}}{1 - \alpha_{t(i)}} \log \left\langle \hat{x}_0\left(x_{t(i)}; \theta\right), x_0 \right\rangle \right]$$

**Multivariate discrete variables.** Now we extend to multivariate discrete data, where we have a sequence of discrete variables. We redefine our notation for this multivariate case $\mathbf{x}_t = \left(x_t^1, x_t^2, \ldots, x_t^L\right)$ where each $x_t^\ell \in \{1, 2, \ldots, K\}$, $L$ is the sequence length, and each position $\ell$ represents a discrete variable in the sequence. The forward process assumes independence across positions:

$$q\left(x_t \mid x_{t-1}\right) = \prod_{\ell=1}^{L} q\left(x_t^\ell \mid x_{t-1}^\ell\right).$$

Each component follows the same absorbing state transition as in the single-variable case: $q\left(x_t^\ell \mid x_{t-1}^\ell\right) = \mathrm{Cat}\left(x_t^\ell;\ \mathbf{e}_{x_{t-1}^\ell}\mathbf{Q}_t\right)$. The reverse process allows for dependencies across positions:

$$p_\theta\left(x_s \mid x_t\right) = \prod_{\ell=1}^{L} p_\theta\left(x_s^\ell \mid x_t^{1:L}\right), \quad s < t.$$

Each position depends on the entire sequence context: $p_\theta\left(x_s^\ell \mid x_t^{1:L}\right) = \mathrm{Cat}\left(x_s^\ell; \mathrm{softmax}\left(f_\theta\left(x_t^{1:L}, t\right)\right)\right)$. Thereby, the training objective extends the single-variable SUBS formulation:

$$\mathcal{L}_{\mathrm{MDLM}}^{\mathrm{multi}} = \sum_{i=1}^{T} \mathbb{E}_q\left[\sum_{\ell=1}^{L} \frac{\alpha_{t(i)} - \alpha_{s(i)}}{1 - \alpha_{t(i)}}\mathbb{I}\left[x_{t(i)}^\ell = [M]\right]\log p_\theta\left(x_0^\ell \mid x_{t(i)}^{1:L}\right)\right],$$

where $\mathbb{I}$ is an indicator function, and it is essential to ensure that we don't compute the loss based on the unmasked tokens.

# H    EXPERIMENTAL DETAILS

## H.1    TEXT-TO-IMAGE DIFFUSION MODELS

**Diffusion model.** Based on Stable Diffusion v1.5 (Rombach et al., 2022), we adopt 50-step DDPM sampling (Ho et al., 2020) with classifier-free guidance (Ho & Salimans, 2021), setting the guidance weight to 5.0 for all experiments. For parameter-efficient fine-tuning, we apply LoRA (Hu et al., 2022) to the denoising UNet with a rank of 4, consistently across all baselines.

**Details of evaluation.** To assess prompt–image alignment, we use CLIPScore (Radford et al., 2021) and ImageReward (Xu et al., 2023). To evaluate diversity, we employ two LPIPS-based metrics (Zhang et al., 2018), which quantify perceptual differences between images. LPIPS-A measures diversity across all generated images, irrespective of the prompt, while LPIPS-P measures diversity within each prompt by computing the mean LPIPS distance among images conditioned on the same prompt. For all evaluations, we sample 32 images per prompt.

**Baselines.** We reproduce results for DDPO[1] (Black et al., 2023) and TDPO[2] (Zhang et al., 2024b) using their official codebases. As the source code for DRaFT is not publicly available, we use a faithful implementation based on the AlignProp codebase[3] (Prabhudesai et al., 2023). We set the batch size to 64 for all fine-tuning methods. For DAS[4] (Kim et al., 2025c), we adjusted its sampling steps from 100 to 50 to match other methods. Accordingly, we set its tempering parameter $\gamma$ to 0.014 to ensure $(1 + \gamma)^T - 1 = 1$ for $T = 50$, following the hyperparameter setting guidelines of the authors.

**Training details of baselines.** Most baselines are trained for 100 epochs. However, due to its slower optimization, we trained DDPO for 500 epochs. For all methods, we report the best performance prior to collapse. In Table 1, the reported results correspond to epoch 400 for DDPO, epoch 40 for DRaFT, and epoch 100 for TDPO. In the case of DAS, we directly report the performance obtained from its test-time inference without additional training.

**Aesthetic score optimization.** We employ the AdamW optimizer (Loshchilov & Hutter, 2019) with a learning rate of 1e-3, $\beta_1 = 0.9$, and $\beta_2 = 0.999$. We set the training batch size to 64. The core hyperparameters for DAV are set as follows: $\alpha = 0.005$, $\gamma = 0.9$, and an importance sampling particle size $M = 4$. For the DAV-KL variant, we use a KL-regularization coefficient of $\lambda = 0.01$. In each iteration, the M-step performs a single update using the dataset collected from the corresponding E-step. We train the model for 100 epochs, which takes approximately 14 hours on eight RTX 3090 (24GB) GPUs. A sensitivity analysis for these hyperparameters is presented in Appendix I.

**Compressibility and incompressibility optimization.** We use the same hyperparameter configuration as the aesthetic score optimization experiment, with the only exception being the importance sampling particle size, which we set to $M = 16$. Training for six epochs takes approximately 2 hours on eight RTX 3090 (24GB) GPUs.

---

[1] https://github.com/kvablack/ddpo-pytorch
[2] https://github.com/ZiyiZhang27/tdpo
[3] https://github.com/mihirp1998/AlignProp
[4] https://github.com/krafton-ai/DAS

## H.2 DNA Sequence Design

**Diffusion Model.** We used the pretrained Masked Diffusion Language Model (MDLM) (Sahoo et al., 2024), following the pretrained diffusion from the Li et al. (2024b). This model is trained on the Enhancer dataset (Gosai et al., 2023), which consists of DNA sequences of length 200.

**Reward Oracle.** Our reward modeling and evaluation setup follows that of DRAKES (Wang et al., 2025). To prevent data leakage, we use two separate reward oracles trained on distinct data splits from Lal et al. (2024). Both oracles are Enformer models (Avsec et al., 2021) initialized with pretrained weights; one serves as the reward signal during fine-tuning, while the other is reserved for held-out evaluation. Further details on the evaluation protocol can be found in Wang et al. (2025), Appendix F.2.

**Evaluation Metrics.** We assess the generated sequences on three criteria: biological validity, naturalness, and diversity. We generate sequences for evaluation using a batch size of 640. To measure biological validity and detect over-optimization, we use an independently trained classifier to predict Chromatin Accessibility (ATAC-Acc) (Lal et al., 2024; Wang et al., 2025; Su et al., 2025). We assess sequence naturalness using the 3-mer Pearson Correlation (3-mer Corr), which compares the k-mer frequencies of generated sequences to those of the top 0.1% most active enhancers in the dataset (Gosai et al., 2023). Finally, we quantify diversity using the average Levenshtein distance between generated sequences (Haldar & Mukhopadhyay, 2011; Kim et al., 2023).

**Predicted activity optimization.** We use the AdamW optimizer (Loshchilov & Hutter, 2019) with a learning rate of 1e-3, $\beta_1 = 0.9$, and $\beta_2 = 0.999$. Hyperparameters for our method are set to $\alpha = 0.01$, $\gamma = 1$, and an importance sampling particle size of $M = 10$. In the M-step, posterior mean approximation at early timesteps is inherently unreliable. Consequently, directly maximizing the likelihood in Equation (10) may compel the model to replicate these erroneous estimates, resulting in unstable optimization. To mitigate this issue, we omit the first 80 steps from the original 128 steps and perform maximum likelihood estimation only on the remaining 48 steps. We train DAV for 200 epochs, which takes approximately 15 hours on a single RTX 3090 (24GB) GPU.

**Experiment Result** Table 2 shows that DAV and its posterior variant, DAV Posterior, achieve a superior balance between reward optimization and validity, and naturalness compared to baseline methods. While strong RL-based methods, such as DDPO and VIDD attain high target rewards, they suffer from a significant drop in diversity and ATAC-acc, which indicates reward over-optimization. In contrast, our amortized DAV policy achieves a higher reward (7.71) than baselines while maintaining high diversity (87.91) and naturalness. Furthermore, the DAV Posterior achieves the highest scores in both the target reward (9.04) and validity (0.865), while still maintaining high diversity. This demonstrates the ability of our methods to generate high-reward sequences without sacrificing naturalness or diversity. For the test-time search baselines (Li et al., 2024b; Chu et al., 2025), although SGDD approaches DAV in Pred-activity and even achieves higher diversity, DAV still substantially outperforms it in ATAC-acc and 3-mer correlation. This demonstrates that DAV Posterior is significantly more robust to reward over-optimization compared to these baselines.

| Method | Pred-activity (↑) (target) | ATAC-acc (↑) (Validity) | 3-mer Corr (↑) (naturalness) | Levenstein Diversity (↑) |
|---|---|---|---|---|
| Pre-trained | 0.13 (0.03) | 0.018 (0.012) | 0.000 (0.081) | **111.58** (0.27) |
| DRAKES | 6.05 (1.09) | 0.784 (0.354) | 0.229 (0.221) | 90.38 (16.20) |
| VIDD | 7.31 (0.09) | 0.545 (0.425) | 0.261 (0.074) | 64.31 (10.06) |
| DDPO | 7.51 (0.05) | 0.129 (0.266) | 0.287 (0.114) | 49.12 (13.83) |
| DAV | 7.71 (0.26) | 0.552 (0.109) | **0.397** (0.145) | 87.91 (3.73) |
| SVDD | 5.06 (0.03) | 0.244 (0.008) | **0.675** (0.004) | 64.72 (0.42) |
| SGDD ($\beta = 30$) | 8.66 (0.04) | 0.225 (0.008) | 0.090 (0.020) | 110.14 (0.07) |
| SGDD ($\beta = 50$) | 8.77 (0.07) | 0.223 (0.021) | 0.090 (0.019) | 109.94 (0.11) |
| DAV Posterior | **9.24 (0.23)** | **0.920 (0.067)** | 0.347 (0.160) | 87.13 (4.63) |

Table 2: Comparison of sequence design methods.

# I  SENSITIVITY TEST

This section provides a sensitivity analysis for the hyperparameters governing the test-time search in DAV. We examine four key parameters: (1) $\alpha$, a temperature parameter that controls the exploration strength, where lower values lead to sharper sampling by amplifying the influence of the soft Q-function; (2) $\gamma$, the discount factor used for credit assignment; (3) *Distillation Steps*, the number of training cycles applied to the dataset collected in a single E-step; and (4) *Number of particles* used for the importance sampling step in Equation (62). The experimental setting, except for these four hyperparameters, follows the setting in Section H.1.

**Temperature $\alpha$.**  As shown in Figure 6, lower $\alpha$ yields higher reward scores. However, excessively small values may cause over-optimization, degrading ImageReward, CLIP score, and diversity. Increasing $\alpha$ mitigates this effect, producing more diverse samples.

**Discount factor $\gamma$.**  Figure 7 shows that setting $\gamma^T \approx 0$ stabilizes optimization. In contrast, high values such as $\gamma = 0.95$ fail to suppress early-step credit assignment, leading to over-optimization. Interestingly, before collapse, reward and alignment scores are positively correlated with $\gamma$, while diversity metrics exhibit a negative correlation.

**Distillation steps.**  As illustrated in Figure 8, increasing the number of distillation steps in the M-step improves the reward when set above one. However, excessive steps reduce both alignment scores and diversity.

**Number of particles.**  Figure 9 shows that the number of particles used for importance sampling has a limited impact on overall performance. Yet, large values can harm alignment scores such as ImageReward and CLIP. For computational efficiency, we set the number of particles to four.

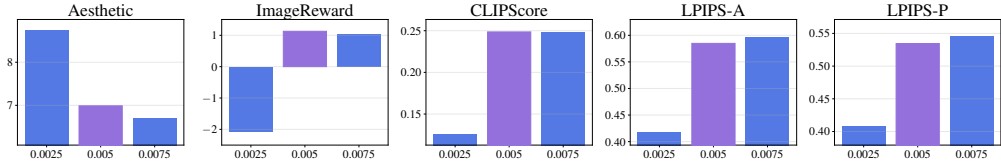

Figure 6: Effect of the inverse temperature parameter $\alpha$ on performance metrics.

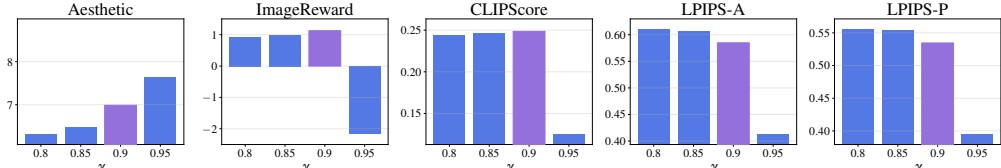

Figure 7: Effect of the discount factor $\gamma$ on performance metrics.

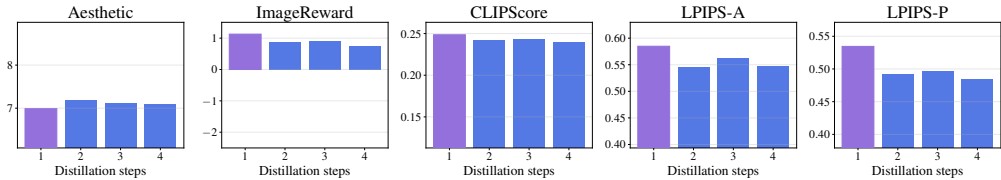

Figure 8: Effect of the number of distillation steps on performance metrics.

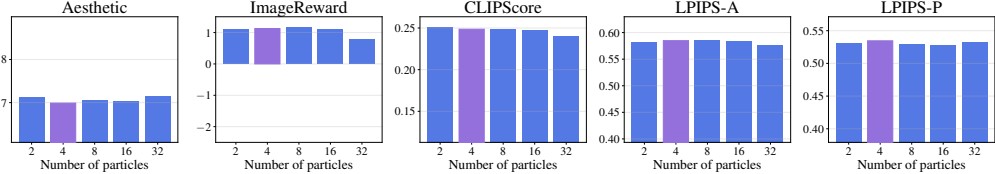

Figure 9: Effect of the number of particles on performance metrics.

## J NON-DIFFERENTIABLE REWARDS

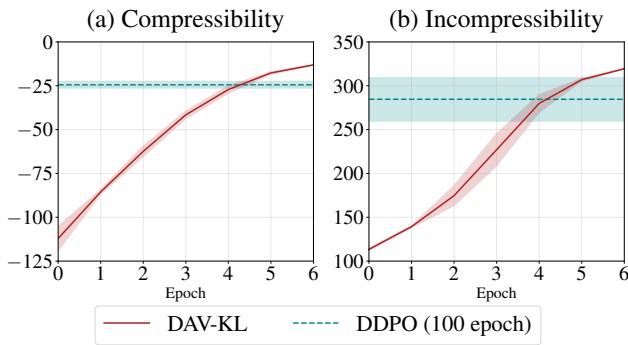

Figure 10: Optimization trends of DAV on compressibility and incompressibility rewards.

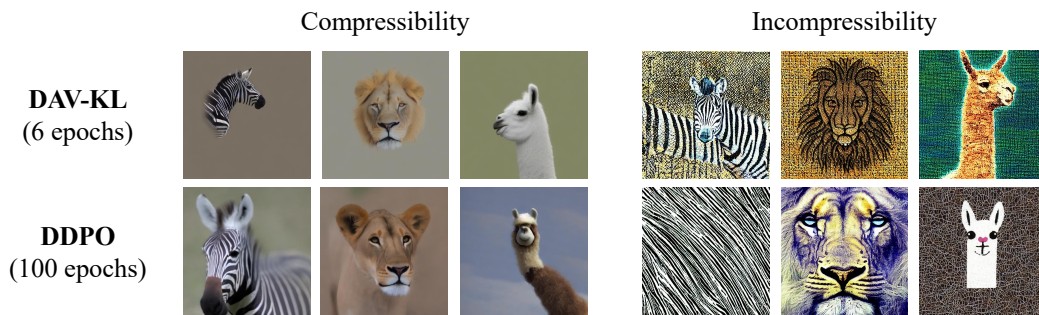

Figure 11: Qualitative comparison between DAV-KL and DDPO when optimized for the compressibility and incompressibility reward function.

DAV extends naturally to non-differentiable rewards by skipping the gradient-based proposal construction described in Section E.2.1. As shown in Figure 10, DAV-KL effectively optimizes both compressibility and incompressibility rewards. This result demonstrates the versatility of DAV in black-box reward optimization.

Figure 11 presents qualitative comparisons between DAV-KL and DDPO. DAV-KL produces more aligned samples with fewer training epochs. Under the compressibility reward, DAV-KL preserves only the core parts of the animals, whereas DDPO retains redundant background textures. For incompressibility, DAV-KL better captures the essential semantic features of the animals, while DDPO often fails to do so despite DAV-KL achieving higher reward scores. These results confirm that DAV is effective for optimizing the non-differentiable rewards.

# K  COMPUTATIONAL COSTS ANALYSIS

**Aesthetic score**  Table 3 reports the RTX 4090 GPU hours and performance of our method compared to DDPO, DRaFT, and their KL-regularized variants. KL-regularized DRAFT and DDPO are optimized following Equation 18 of Uehara et al. (2024a):

$$p^* = \arg\max_{p_\theta} \mathbb{E}_{\tau \sim p_\theta(\tau)} \left[ r(x_0) - \alpha \sum_{t=1}^{T} \mathcal{D}_{KL}(p_\theta(\cdot \mid x_t) || p_{\theta^0}(\cdot \mid x_t)) \right].$$

While DAV requires a substantial computational budget, its runtime remains comparable to the high-epoch DDPO and KL-regularized baselines. Crucially, DAV justifies this cost by achieving a superior trade-off: it attains the highest aesthetic scores while preserving LPIPS-A and ImageReward. In contrast, KL-regularized baselines suffer significant degradation in diversity and ImageReward even when consuming comparable or greater GPU hours.

| Method-epochs | Aesthetic (↑) | LPIPS-A (↑) | ImageReward (↑) | GPU hours |
|---|---|---|---|---|
| Pretrained | 5.40 | **0.65** | 0.90 | - |
| DDPO-100 | 6.08 | 0.63 | 0.96 | 18.1 |
| DDPO-200 | 6.44 | 0.57 | 0.85 | 36.1 |
| DDPO-300 | 6.70 | 0.54 | 0.67 | 54.2 |
| DDPO-400 | 6.84 | 0.48 | 0.28 | 72.2 |
| DDPO-500 | 6.82 | 0.44 | -0.44 | 90.3 |
| DDPO+KL-400 ($\alpha$=0.3) | 6.93 | 0.47 | 0.47 | 82.7 |
| DRaFT-42 | 7.22 | 0.46 | 0.19 | **1.7** |
| DRaFT+KL-2000 ($\alpha$=0.035) | 6.78 | 0.59 | 0.23 | 220.0 |
| DAV-100 (M=4) | **8.04** | 0.53 | 0.95 | 82.4 |
| DAV-KL-100 (M=2) | 7.11 | 0.58 | 1.11 | 91.2 |
| DAV-KL-100 (M=4) | 6.99 | 0.58 | **1.13** | 98.7 |

Table 3: Comparison of computational cost and performance for optimizing aesthetic score.

**Compressibility and incompressibility**  As shown in Figure 10, DAV-KL trained for 6 epochs substantially outperforms the DDPO baseline trained for 100 epochs. In terms of compute, DAV-KL requires 14.3 GPU hours on an RTX 3090, which is roughly half the cost of DDPO at 28.7 GPU hours.

**DNA sequence design**  For discrete diffusion model fine-tuning, we reproduce DDPO (Black et al., 2023) and VIDD (Su et al., 2025) using the official codebases of VIDD [5], and we reproduce DRAKES (Wang et al., 2025) following its official implementation [6]. All hyperparameters are set exactly as specified in the original papers. On a single RTX 3090 GPU, the training times are approximately: 14 hours for DDPO, 16 hours for VIDD, 43 hours for DRAKES, and 15 hours for DAV. Notably, DAV achieves comparable training time while yielding higher reward and naturalness with preserved diversity.

---

[5] https://github.com/divelab/VIDD
[6] https://github.com/ChenyuWang-Monica/DRAKES

