# OpenReview forum: "Diffusion Alignment as Variational Expectation-Maximization"
_ICLR.cc/2026/Conference — ICLR 2026 Poster_

### Official Review · Reviewer_xurq · 2025-10-28

**Soundness:** 3
**Presentation:** 3
**Contribution:** 3
**Rating:** 6
**Confidence:** 4

**Summary:**

The paper develops an approach to “Diffusion Alignment as Variational Expectation-Maximization” (DAV) that alternates between two complementary phases: the E-step, which is essentially an exploration step that aims to discover diverse and high-reward samples from the variational posterior; and the M-step, “amortization”,  which refines the diffusion model using samples identified from the E-step.

**Strengths:**

The DAV approach is built on solid technical foundations as outlined in \S4. For instance, the E-step uses gradient-based guidance and importance sample to enhance the exploration. The M-step minimizes a mode-covering objective that incentivizes the covering of all diverse modes generated in the E-step.

The combined E-M steps iteratively refine the model towards a multi-modal aligned distribution; and this overcomes problems like over-optimization and mode collapse that often arise in RL.

**Weaknesses:**

The M-step involves two variations, in addition to the standard DAV objective in (10), there’s a variation,  DAV-KL in (11). From the experimental results in Table 1, it’s not clear which one to use and when, except perhaps the ad hoc trial and error.

**Questions:**

Can the author(s) shed some light on how to choose the value of the KL-coefficient \lambda in (11), which is meant to control “the trade-off between aligning with the expert policy and pre-serving the pretrained model”? In particular, is the value \lambda robust or not with respect to downstream applications?

---

> ### Author Response · Authors · 2025-11-21
>
> > **(Weakness 1, Question 1)** Can the author(s) shed some light on how to choose the value of the KL-coefficient \lambda in (11), which is meant to control “the trade-off between aligning with the expert policy and preserving the pretrained model”?
>
> We thank the reviewer for raising this question. Table 1 provides a sensitivity analysis of the KL-coefficient $\lambda$. As $\lambda$ increases, the aesthetic score gradually decreases while diversity (LPIPS-A) increases, eventually approaching the pretrained model’s characteristics. This behavior aligns with the intended interpretation of $\lambda$:
>
> - Smaller $\lambda$ → stronger alignment, which leads to a higher aesthetic score and reduced diversity.
> - Larger $\lambda$ → stronger preservation of the pretrained model for higher diversity and slightly lower reward.
>
> The trend is smooth and predictable, with nearly linear changes across $\lambda$ values, making the trade-off easy to control in practice. So, we recommend starting with a small $\lambda$, such as 0.01 or 0.001, and adjusting it based on your target: whether to gain more reward-aligned samples or preserve diversity. We use $\lambda=0.01$ by default for DAV-KL.
>
> Table 1. Sensitivity test of KL-coefficient $\lambda$
>
> |                  | Aesthetic (↑) | LPIPS-A (↑) |
> |------------------|---------------|-------------|
> | KL=0.00 (DAV)    | 8.04          | 0.53        |
> | KL=0.005         | 7.49          | 0.55        |
> | KL=0.01          | 6.99          | 0.58        |
> | KL=0.015         | 6.52          | 0.61        |
> | Pretrained       | 5.40          | 0.65        |

---

### Official Review · Reviewer_KMQx · 2025-10-30

**Soundness:** 2
**Presentation:** 3
**Contribution:** 2
**Rating:** 6
**Confidence:** 4

**Summary:**

The paper introduces DAV, a novel framework for fine-tuning pre-trained diffusion models. The authors motivate the work by claiming to address reward over-optimization.

**Strengths:**

1. This work formulates diffusion alignment as an iterative Variational Expectation-Maximization, which appears to be a new and interesting theoretical lens for diffusion fine-tuning.2.
2. The proposed DAV framework enjoys broad applicability. It can accommodate both continuous and discrete settings.
3. The presentation of this work is easy to follow.
4. The empirical results show DAV and DAV-KL enjoy superiority over multiple strong baselines, such as DDPO, DRaFT.

**Weaknesses:**

1.  The main comparison in Figure 3, which plots aesthetic reward against diversity/naturalness, is confusing and potentially incomplete. The reported performance of the RL-based if they are properly trained with "suitable" KL penalty (which might be non-trivial to choose). This raises questions about the optimal tuning of these baselines.
2. Furthermore, the analysis in Figure 3 omits purely inference-time methods, which are often competitive in image experiments.
3. As noticed by the authors, this method has non-negligible computation costs. The E-step involves substantial "additional test-time computation" through gradient-guided search. In large-scale diffusion model finetuning, DDPO already takes much time to converge (compared to the fastest direct propagation). It is important to quantitatively present the added training-time overhead of DAV relative to DDPO.

**Questions:**

1. For the results presented in Figure 3, do the images for each algorithm come from a single training run, or are they gathered over multiple runs? If from a single run, please report the standard deviation rather than only the mean.
2. For discrete finetuning, usually it's straightforward to test both DNA and RNA sequences. Can the authors provide explanations on why only the DNA enhancer is tested?
3. See other questions above

---

> ### Author Response · Authors · 2025-11-21
>
> > **(Weakness 1)** The reported performance of the RL-based if they are properly trained with "suitable" KL penalty (which might be non-trivial to choose). This raises questions about the optimal tuning of these baselines.
>
> Thank you for the constructive feedback. We agree that a clearer discussion of baselines with a suitable KL penalty is valuable. We investigated this by penalizing the objective function of DRaFT and DDPO with the KL divergence multiplied by
>
> \begin{align}
> p^*(x) = \arg\max \_{p\_{\_\theta}} \mathbb E\_{\tau \sim p\_{\theta}}\left[r(x\_0)-\alpha \sum\_{t=1}^T D\_\text{KL}(p\_\theta(\cdot|x\_t) || p\_{\theta^0}(\cdot|x\_t) )\right]
> \end{align}
>
> which is a widely adopted strategy for KL-regularized diffusion finetuning. For more details, see equation 18 of [1].
>
> The results in Table 1 show that even with KL regularization, the baselines struggle to mitigate over-optimization. For example, while DRaFT+KL shows it can preserve diversity (LPIPS-A), its aesthetic score and ImageReward drop significantly. In contrast, our DAV methods achieve high reward scores and ImageReward while maintaining strong diversity.
>
> Table 1. DAV vs. KL-penalized baselines, averaged over 3 seeds.
> |   | Aesthetic (↑) | LPIPS-A (↑) | ImageReward (↑) |
> |--|---|--|--|
> | DDPO+KL-400 ($\alpha$=0.4) | 6.76  | -0.02 | 0.50|
> | DDPO+KL-400 ($\alpha$=0.3) | 6.93   | 0.47 | 0.47 |
> | DDPO+KL-400 ($\alpha$=0.2) | 6.92 | -1.04 | 0.40|
> | DRaFT+KL ($\alpha$=0.04)   | 6.69  | **0.60** | 0.55  |
> | DRaFT+KL ($\alpha$=0.035)  | 6.78   | 0.59 | 0.23   |
> | DRaFT+KL ($\alpha$=0.03)   | 7.80  | 0.52 | -1.50 |
> | DAV-100 | **8.04**  | 0.53   | 0.95 |
> | DAV-KL-100  | 6.99  | 0.58   | **1.13**|
>
> >**(Weakness 2)** Furthermore, the analysis in Figure 3 omits purely inference-time methods, which are often competitive in image experiments.
>
> In Figure 3 of the paper, we already include one of the most competitive inference-time methods, DAS [2], as a key baseline (fourth row). Its quantitative results are also reported in Table 1 of the paper. As shown in the table, DAV-KL-Posterior (finetuned model w/ inference-time computation) outperforms DAS in both aesthetic score (7.22 → 8.66) and ImageReward score (1.07 → 1.11), while losing diversity slightly (0.65 → 0.58). Moreover, DAV w/o inference-time computation also surpasses DAS in aesthetic quality (7.22 → 8.04).
>
> > **(Weakness 3)** It is important to quantitatively present the added training-time overhead of DAV relative to DDPO.
>
> We agree with your comment, and we have included a detailed analysis of runtime and training costs in the official comment. In addition to the DDPO comparison you pointed out, we further provide analysis of computation cost against KL-regularized variants of baselines. This analysis shows that DAV consistently achieves competitive or superior computational efficiency on all tasks. The full results are available in the official comment.
>
> > **(Question 1)** For the results presented in Figure 3, do the images for each algorithm come from a single training run, or are they gathered over multiple runs? If from a single run, please report the standard deviation rather than only the mean.
>
> We apologize for the lack of clarity regarding how the qualitative results in Figure 3 were selected. All images in Figure 3 were collected from multiple training runs, and the reward value displayed under each method corresponds to the average reward of the seven sampled images shown in the figure. We clarify this by adding these details to the caption of Figure 3.
>
> > **(Question 2)** Can the authors provide explanations on why only the DNA enhancer is tested?
>
> For the DNA enhancer task, Chromatin Accessibility (ATAC-Acc) [3,4] is a well-established method for validating generated DNA sequences. In our work, reliably detecting the overoptimization is important, and ATAC-Acc is a key property for detecting over-optimization. Previous works validate their methods via ATAC-Acc [5,6,7], and open-sourced codebases for reproducing the whole benchmark for the enhancer task are available.
>
> As you noted, 5'UTR optimization is an important and active research area. However, unlike the DNA enhancer benchmark, a reproducible and reliable evaluation framework for 5'UTR with a reliable detector for over-optimization is still lacking. For this reason, we focus our experiments on the DNA enhancer setting.

---

> ### Author Response · Authors · 2025-11-21
>
> [1] Uehara, Masatoshi, et al. "Understanding reinforcement learning-based fine-tuning of diffusion models: A tutorial and review." arXiv preprint arXiv:2407.13734 (2024).
>
> [2] Kim, Sunwoo, Minkyu Kim, and Dongmin Park. "Test-time alignment of diffusion models without reward over-optimization." arXiv preprint arXiv:2501.05803 (2025).
>
> [3] ENCODE Project Consortium. "An integrated encyclopedia of DNA elements in the human genome." Nature 489.7414 (2012): 57.
>
> [4] Lal, Avantika, et al. "Designing realistic regulatory DNA with autoregressive language models." Genome Research 34.9 (2024): 1411-1420.
>
> [5] Su, Xingyu, et al. "Iterative Distillation for Reward-Guided Fine-Tuning of Diffusion Models in Biomolecular Design." arXiv preprint arXiv:2507.00445 (2025).
>
> [6] Wang, Chenyu, et al. "Fine-Tuning Discrete Diffusion Models via Reward Optimization with Applications to DNA and Protein Design." The Thirteenth International Conference on Learning Representations.
>
> [7] Chu, Wenda, et al. "Split gibbs discrete diffusion posterior sampling." arXiv preprint arXiv:2503.01161 (2025).

---

### Official Review · Reviewer_77LS · 2025-11-01

**Soundness:** 3
**Presentation:** 3
**Contribution:** 2
**Rating:** 6
**Confidence:** 4

**Summary:**

The authors propose DAV, an alignment algorithm for diffusion models via variational expectation maximization. In the E-step, DAV employs test-time search algorithms to generate samples from the reward-weighted posterior distribution. In the M-step, the diffusion model is fine-tuned using the samples from the E-step. The authors demonstrate the effectiveness of DAV on both the continuous text-to-image generation task and the discrete DNA sequence design.

**Strengths:**

- The idea to formulate diffusion alignment as a variational expectation-maximization problem is interesting.
- The paper is well-written, and the theory and method are well-motivated.
- Experiments showcase the effectiveness of DAV.

**Weaknesses:**

- The searching algorithms lead to computational overhead. Therefore, a fairer comparison with the baselines should also take the computational cost into account. For example, it would be helpful to compare model performance under the same computational budget and the performance scaling curve of TR2-D2 as the computation increases.
- The value of 3-mer correlation for DNA sequence design is significantly lower than those reported by baselines, e.g., in DRAKES paper the value is 0.887, much higher than DAV (0.397), while in table 2 and figure 5, it is only 0.229. Also, the target and naturalness are two competing properties, and one can get a higher value of one property by sacrificing the other via hyperparam tuning or using different training epochs. Does DAV have Pareto optimal performance compared to baselines?
- The E-step can lead to an inaccurate estimation of the posterior distribution, due to the limited sample size and the value estimation error in the test-time algorithms. How does this affect the M-step optimization, and is the model robust to a suboptimal posterior distribution (e.g., with fewer samples or inaccurate samples)?

**Questions:**

Please refer to the **weaknesses** section.

---

> ### Author Response · Authors · 2025-11-21
>
> >**(Weakness 1)** Therefore, a fairer comparison with the baselines should also take the computational cost into account.
>
> In our official comment, we now include a comprehensive computational cost analysis across all three major tasks: aesthetic optimization, compressibility & incompressibility metrics, and DNA enhancer activity. The results show that DAV achieves strong reward optimization while avoiding over-optimization under comparable or lower GPU budgets. We kindly refer the reviewer to the official comment for a more detailed comparison.
>
>
> >**(Weakness 2)** The value of 3-mer correlation for DNA sequence design is significantly lower than those reported by baselines, e.g., in DRAKES paper the value is 0.887, much higher than DAV (0.397), while in Table 2 and Figure 5, it is only 0.229. Also, the target and naturalness are two competing properties, and one can get a higher value of one property by sacrificing the other via hyperparameter tuning or using different training epochs. Does DAV have Pareto optimal performance compared to baselines?
>
> First, we clarify that "DRAKES" in our main paper refers to "DRAKES w/o KL," following the naming convention in the VIDD paper [1]. For completeness and clarity, we have added "DRAKES w/ KL" from the DRAKES paper [2] to Table 1 below.
>
> To analyze the trade-off between target reward (Pred-activity) and naturalness (3-mer Corr) in DAV, we varied the particle number for importance sampling as $M\in\{5,10,20\}$. As shown in Table 1, while DRAKES w/ KL achieves the highest naturalness, removing the KL constraint results in a significant degradation in naturalness (0.887 $\to$ 0.229) for only a marginal gain in reward (5.61 $\to$ 6.05). In contrast, DAV demonstrates a superior and smoother trade-off: increasing $M$ gradually improves reward while decreasing naturalness in a more controlled manner.
>
> Table 1. Effect of the $M$ on the reward–naturalness trade-off.
> | Method       | Pred-activity ↑ (target) | 3-mer Corr ↑ (naturalness) |
> | ------------ | ------------------------ | -------------------------- |
> | Pre-trained  | 0.13 (0.03)              | 0.000 (0.081)              |
> | DRAKES w/ KL | 5.61 (0.07)              | 0.887 (0.002)              |
> | DRAKES w/o KL| 6.05 (1.09)              | 0.229 (0.221)              |
> | VIDD         | 7.31 (0.09)              | 0.261 (0.074)              |
> | DDPO         | 7.51 (0.05)              | 0.287 (0.114)              |
> | DAV ($M=5$)  | 7.03 (0.26)              | 0.607 (0.011)              |
> | DAV ($M=10$) | 7.71 (0.26)              | 0.397 (0.145)              |
> | DAV ($M=20$) | **8.11 (0.29)**          | 0.042 (0.150)              |
>
> [1] Su, Xingyu, et al. "Iterative Distillation for Reward-Guided Fine-Tuning of Diffusion Models in Biomolecular Design." arXiv preprint arXiv:2507.00445 (2025).
>
> [2] Wang, Chenyu, et al. "Fine-Tuning Discrete Diffusion Models via Reward Optimization with Applications to DNA and Protein Design." The Thirteenth International Conference on Learning Representations.
>
> >**(Weakness 3)** The E-step can lead to an inaccurate estimation of the posterior distribution, due to the limited sample size and the value estimation error in the test-time algorithms. How does this affect the M-step optimization, and is the model robust to a suboptimal posterior distribution (e.g., with fewer samples or inaccurate samples)?
>
> As you pointed out, the E-step may yield an inaccurate posterior due to limited samples or value estimation error. However, we empirically show that DAV successfully optimizes rewards while mitigating over-optimization across four reward functions and both continuous and discrete diffusion models, even in the presence of such error. Empirically, the sensitivity analysis in Appendix J shows that DAV is robust to the number of particles used in the importance sampling step, indicating that the M-step optimization remains stable even when the posterior distribution is estimated with fewer or noisier samples.

---

### Official Review · Reviewer_8DWZ · 2025-11-01

**Soundness:** 2
**Presentation:** 2
**Contribution:** 3
**Rating:** 6
**Confidence:** 3

**Summary:**

This paper presents a diffusion alignment method (DAV) that alternates between test-time searches as Expectation steps and online refinement of the diffusion model as Maximization steps. Specifically, the diffusion alignment problem is formulated as a soft RL objective, whose evidence lower bound is optimized an EM algorithm. In the E-steps, DAV draws posterior samples given a reward-tilted distribution; while in the M-steps, DAV distills the sampled trajectories into the diffusion model. Experimental results show the effectiveness of DAV compared to existing RL and direct preference optimization methods for both continuous and discrete diffusion models.

**Strengths:**

- The paper offers a fresh perspective by aligning diffusion models with the EM algorithm. I especially appreciate this idea because the multi-round iterative alignment could potentially help in settings where the reward is costly or intractable to evaluate—for example, when it requires human evaluation or expensive wet-lab experiments.
- The proposed method is accompanied by rigorous derivations and theoretical guarantees.

**Weaknesses:**

- Experiments only include one example for continuous diffusion and one for discrete diffusion. The case for generality would be stronger with additional tasks (e.g., compressibility or prompt alignment as in DDPO). Moreover, some of the recent methods are not included or discussed as well, such as DSPO[1], DanceGRPO[2].
- The EM algorithm may be substantially more expensive than the methods it is compared to (e.g., DRaFT or DDPO), given the test-time search required in each expectation step. However, there is currently no analysis on the runtime or convergence speed of DAV.

[1] Zhu et al. "DSPO: Direct Score Preference Optimization for Diffusion Model Alignment", ICLR 2025.

[2] Xue et al. "DanceGRPO: Unleashing GRPO on Visual Generation", arXiv: 2505.07818.

**Questions:**

- For the discrete diffusion model alignment, I am curious of how DAV compare to test-time sampling algorithms such as [3,4], which also consider the same DNA enhancer task, and also alignment methods designed for discrete diffusion models (e.g., [5,6]).

[3] Li et al. "Derivative-Free Guidance in Continuous and Discrete Diffusion Models with Soft Value-Based Decoding", arXiv: 2408.08252.

[4] Chu et al. "Split Gibbs Discrete Diffusion Posterior Sampling", NeurIPS 2025.

[5] Borso et al. "Preference-Based Alignment of Discrete Diffusion Models", ICLR 2025 Bi-Align Workshop.

[6] Zhu et al. "LLaDA 1.5: Variance-Reduced Preference Optimization for Large Language Diffusion Models", arXiv: 2505.19223.

---

> ### Author Response · Authors · 2025-11-21
>
> >**(Weakness 1-1)** The case for generality would be stronger with additional tasks (e.g., compressibility or prompt alignment as in DDPO).
>
> Since DAV is built upon the general probabilistic inference framework, DAV can optimize the diffusion model for arbitrary reward functions, regardless of their differentiability. In Appendix J, we already report the result of DAV to optimize the compressibility and incompressibility score from DDPO [1]. As shown in Figure 10, DAV-KL significantly outperforms the DDPO. In terms of compute, DAV-KL required 14.3 GPU hours of RTX 3090, which is approximately half the computational time of DDPO at 28.7 hours.
>
> >**(Weakness 1-2)** recent methods are not included or discussed as well, such as DSPO[1], DanceGRPO[2]
>
> Thank you for your comment, which gave us the opportunity to clarify how our work relates to prior research. We have incorporated additional related works into Appendix F.2 of the revised manuscript.
>
> We notice a recent surge in GRPO-based methods for T2I models, such as Flow-GRPO [2] and DanceGRPO [3]. While GRPO-based approaches demonstrate strong optimization performance, their primary focus is maximizing reward rather than mitigating over-optimization. In contrast, DAV is designed to mitigate over-optimization, balancing reward optimization with diversity and alignment score.
>
> In the case of DPO-based alignment methods, including DSPO [4] and Diffusion-KTO [5], they achieve a high human-preference win rate over pretrained models. However, they are constrained by pre-collected preference datasets, which limit their upper bound on performance [3]. Furthermore, while prior GRPO and DPO-based works focus primarily on visual generative tasks, DAV offers a unified framework that is empirically validated on both continuous and discrete diffusion models.
>
> >**(Weakness 2)** there is currently no analysis on the runtime or convergence speed of DAV.
>
> To address the lack of detailed runtime and computation cost analysis, we provide a comprehensive computational cost analysis in our official comment. The results show that DAV effectively optimizes the reward while mitigating over-optimization within a comparable computational budget for the three core tasks: aesthetic score, compressibility & incompressibility score, and DNA enhancer activity. For the full comparison, please refer to the official comment.
>
>
> >**(Question 1)**  I am curious of how DAV compare to test-time sampling algorithms such as [6,7], which also consider the same DNA enhancer task, and also alignment methods designed for discrete diffusion models (e.g., [8,9]).
>
> Table 1 below compares DAV with recent test-time search algorithms [6][7] on DNA enhancer tasks. The results demonstrate that DAV Posterior achieves the highest Pred-activity and ATAC-acc scores while preserving naturalness and diversity. Although SGDD approaches DAV in terms of Pred-activity and offers higher diversity, DAV significantly outperforms it in ATAC-acc and 3-mer correlation. This indicates that DAV Posterior is more robust against reward over-optimization compared to the baselines. Please note that the ATAC-acc result at the [7] is based on a specific evaluation setting. Unlike the original SGDD paper, which evaluated solely on the HepG2 cell line, we adhered to the established benchmark by measuring the average chromatin accessibility across all cell lines. Consequently, this methodological difference accounts for the discrepancy between the ATAC-acc values reported in the SGDD paper and our reproduced results.
>
>
> Table 1. Comparison of DAV Posterior with test-time search baselines.
> | Method        | Pred-activity ↑ (target) | ATAC-acc ↑ (Validity) | 3-mer Corr ↑ (naturalness) | Levenstein Diversity ↑ |
> |---------------|---------------------------|-------------------------|-----------------------------|--------------------------|
> | Pre-trained   | 0.13 (0.03)               | 0.018 (0.012)          | 0.000 (0.081)               | **111.58 (0.27)**        |
> |SVDD|5.06 (0.03)|0.244 (0.008)|**0.675 (0.004)**|64.72 (0.42)|
> |SGDD ($\beta=30$)|8.66 (0.04)| 0.225 (0.008)|0.090 (0.020)|110.14 (0.07)|
> |SGDD ($\beta=50$)|8.77 (0.07)| 0.223 (0.021)|0.090 (0.019)|109.94 (0.11)
> | DAV Posterior | **9.24 (0.23)**           | **0.920 (0.067)**      | 0.347 (0.160)               | 87.13 (4.63)             |
>
>
> In the case of DPO-based discrete diffusion alignment methods, they inherit the same limitation of their continuous diffusion counterparts [4][5], making online RL approaches like DAV more advantageous. Furthermore, regrettably, the source code of [8][9] is not publicly available, preventing a faithful reproduction in a limited time.

---

> ### Author Response · Authors · 2025-11-21
>
> [1] Black, Kevin, et al. "Training Diffusion Models with Reinforcement Learning." The Twelfth International Conference on Learning Representations.
>
> [2] Liu, Jie, et al. "Flow-grpo: Training flow matching models via online rl." arXiv preprint arXiv:2505.05470 (2025).
>
> [3] Xue, Zeyue, et al. "DanceGRPO: Unleashing GRPO on Visual Generation." arXiv preprint arXiv:2505.07818 (2025).
>
> [4] Zhu, Huaisheng, Teng Xiao, and Vasant G. Honavar. "DSPO: Direct score preference optimization for diffusion model alignment." The Thirteenth International Conference on Learning Representations. 2025.
>
> [5] Li, Shufan, et al. "Aligning diffusion models by optimizing human utility." Advances in Neural Information Processing Systems 37 (2024): 24897-24925.
>
> [6] Li, Xiner, et al. "Derivative-free guidance in continuous and discrete diffusion models with soft value-based decoding." arXiv preprint arXiv:2408.08252 (2024).
>
> [7] Chu, Wenda, et al. "Split gibbs discrete diffusion posterior sampling." arXiv preprint arXiv:2503.01161 (2025).
>
> [8] Borso et al. "Preference-Based Alignment of Discrete Diffusion Models", ICLR 2025 Bi-Align Workshop.
>
> [9] Zhu, Fengqi, et al. "LLaDA 1.5: Variance-Reduced Preference Optimization for Large Language Diffusion Models." arXiv preprint arXiv:2505.19223 (2025).

---

### Author Response · Authors · 2025-11-21

We sincerely thank the review committee for their thoughtful and constructive feedback. We appreciate the recognition of our paper’s strengths, simultaneously emphasized by the reviewers: **Novelty** (8DWZ, KMQx), **Technically solid** (8DWZ, xurq), **Strong experimental results** (77LS, KMQx), and **Clear writing and motivation** (77LS, KMQz).

We also acknowledge the shared concern regarding computational expense raised by (M6wy, 77LS, KMQx). To address this concern, we now provide detailed computational costs analysis for optimizing (1) aesthetic score, (2) compressibility & incompressibility metrics, and (3) DNA enhancer activity.

---
**Computational costs analysis for aesthetic score**

Table A reports the RTX 4090 GPU hours and performance of our method compared to DDPO, DRaFT, and their KL-regularized variants following Eq. 18 of [1]. While DAV requires a substantial computational budget, its runtime remains comparable to the high-epoch DDPO and KL-regularized baselines. Crucially, DAV justifies this cost by achieving a superior trade-off: it attains the highest aesthetic scores while preserving LPIPS-A and ImageReward. In contrast, KL-regularized baselines suffer significant degradation in diversity and ImageReward even when consuming comparable or greater GPU hours.


**Table A.** Comparison of computational cost and performance.
| Method-(epochs)           | Aesthetic (↑) | LPIPS-A (↑) | ImageReward (↑) | GPU hours |
|:----------:|:-------------:|:-----------:|-----------------|-----------|
| Pretrained       | 5.40          | **0.65**       | 0.90 | -
| DDPO-100   | 6.08          | 0.63        | 0.96            | 18.1      |
| DDPO-200   | 6.44          | 0.57        | 0.85            | 36.1     |
| DDPO-300   | 6.70          | 0.54        | 0.67            | 54.2      |
| DDPO-400   | 6.84          | 0.48        | 0.28            | 72.2    |
| DDPO-500   | 6.82          | 0.44        | -0.44           | 90.3     |
| DDPO+KL-400 ($\alpha$=0.3) | 6.93          | 0.47        | 0.47            |  82.7         |
|DRaFT-42 |7.22|0.46|0.19| **1.7**|
| DRaFT+KL-2000 ($\alpha$=0.035)  | 6.78          | 0.59        | 0.23            | 220.0 |
| DAV-100 (M=4)          | **8.04**          | 0.53        | 0.95            | 82.4|
| DAV-KL-100 (M=2) | 7.11      | 0.58    | 1.11        | 91.2     |
| DAV-KL-100 (M=4) | 6.99      | 0.58    | **1.13**        | 98.7     |


---
**Computational costs analysis for compressibility and incompressibility**

We note that the analysis of compressibility and incompressibility score [2] was already included in the original paper in Appendix J. As shown in Figure 10, DAV-KL trained for 6 epochs substantially outperforms the DDPO baseline trained for 100 epochs. In terms of compute, DAV-KL requires 14.3 GPU hours on an RTX 3090, which is roughly half the cost of DDPO at 28.7 GPU hours.

---
**Computational costs analysis for DNA sequence design**

For discrete diffusion model fine-tuning, we reproduce DDPO [2] and VIDD [3] using the [official codebases of VIDD](https://github.com/divelab/VIDD), and we reproduce DRAKES [4] following its [official implementation](https://github.com/ChenyuWang-Monica/DRAKES). All hyperparameters are set exactly as specified in the original papers. On a single RTX 3090 GPU, the training times are approximately: **14 hours for DDPO, 16 hours for VIDD, 43 hours for DRAKES, and 15 hours for DAV**. Notably, DAV achieves comparable training time while yielding higher reward and naturalness with preserved diversity as presented our paper.

[1] Uehara, Masatoshi, et al. "Understanding reinforcement learning-based fine-tuning of diffusion models: A tutorial and review." arXiv preprint arXiv:2407.13734 (2024).

[2] Black, Kevin, et al. "Training Diffusion Models with Reinforcement Learning." The Twelfth International Conference on Learning Representations.

[3] Su, Xingyu, et al. "Iterative Distillation for Reward-Guided Fine-Tuning of Diffusion Models in Biomolecular Design." arXiv preprint arXiv:2507.00445 (2025).

[4] Wang, Chenyu, et al. "Fine-Tuning Discrete Diffusion Models via Reward Optimization with Applications to DNA and Protein Design." The Thirteenth International Conference on Learning Representations.

---

### Author Response · Authors · 2025-11-21
**Revision notification**

To address the reviewers’ feedback, we have revised our manuscript and highlighted the corresponding changes in red.

- Figure 3: Following the clarity concern raised by KMQx, we revise the caption to provide a more precise explanation.
- Appendix F.2: We add a discussion of recent GRPO and DPO-based approaches to better situate our work within the current literature.
- Appendix H.2: We incorporated additional test-time search baselines for DNA sequence design.
- Appendix K: We included a detailed computational cost analysis.

---

### Author Response · Authors · 2025-11-28

Dear Reviewers,

We hope this message finds you well.

We have noticed that reviewers have not yet participated in the discussion. As the author-reviewer discussion period is set to conclude in a few days, we kindly wish to remind you of the opportunity to provide your valuable feedback.

Your insights and perspectives are incredibly important to us and will greatly contribute to improving our work. If you have any questions or require further information, please do not hesitate to reach out.

We sincerely appreciate your time and consideration, and we look forward to your input.

Best regards, The Authors

---

### Meta-Review · Area_Chair_DBte · 2026-01-04

**Summary:**

This paper presents a diffusion alignment method (DAV) from a Variational Expectation-Maximization perspective that alternates between test-time searches as Expectation steps and online refinement of the diffusion model as Maximization steps.

The Reviewer Scores consistently reach above the acceptance threshold.
The main concerns of reviewers focus on computational overhead and the insufficient comparison tasks in experiments.
The authors provide additional computational cost comparisons and address the reviewers' concerns.

**Reviewer Concerns:**

The main concerns of reviewers focus on computational overhead and the insufficient comparison tasks in experiments.

The authors provide additional computational cost comparisons.

**Reviewer Scores:**

The Reviewer Scores consistently reach above the acceptance threshold.

---

### Decision · Program_Chairs · 2026-01-26

Accept (Poster)